# Consistent Robust Regression

**Kush Bhatia**[*]
University of California, Berkeley
kushbhatia@berkeley.edu

**Prateek Jain**
Microsoft Research, India
prajain@microsoft.com

**Parameswaran Kamalaruban**[†]
EPFL, Switzerland
kamalaruban.parameswaran@epfl.ch

**Purushottam Kar**
Indian Institute of Technology, Kanpur
purushot@cse.iitk.ac.in

## Abstract

We present the first efficient and provably consistent estimator for the robust regression problem. The area of robust learning and optimization has generated a significant amount of interest in the learning and statistics communities in recent years owing to its applicability in scenarios with corrupted data, as well as in handling model mis-specifications. In particular, special interest has been devoted to the fundamental problem of robust linear regression where estimators that can tolerate corruption in up to a constant fraction of the response variables are widely studied. Surprisingly however, to this date, we are not aware of a polynomial time estimator that offers a consistent estimate in the presence of dense, unbounded corruptions. In this work we present such an estimator, called CRR. This solves an open problem put forward in the work of [3]. Our consistency analysis requires a novel two-stage proof technique involving a careful analysis of the stability of ordered lists which may be of independent interest. We show that CRR not only offers consistent estimates, but is empirically far superior to several other recently proposed algorithms for the robust regression problem, including *extended Lasso* and the TORRENT algorithm. In comparison, CRR offers comparable or better model recovery but with runtimes that are faster by an order of magnitude.

## 1 Introduction

The problem of robust learning involves designing and analyzing learning algorithms that can extract the underlying model despite dense, possibly malicious, corruptions in the training data provided to the algorithm. The problem has been studied in a dizzying variety of models and settings ranging from regression [19], classification [11], dimensionality reduction [4] and matrix completion [8].

In this paper we are interested in the Robust Least Squares Regression (RLSR) problem that finds several applications to robust methods in face recognition and vision [22, 21], and economics [19]. In this problem, we are given a set of $n$ covariates in $d$ dimensions, arranged as a data matrix $X = [\mathbf{x}_1, \ldots, \mathbf{x}_n]$, and a response vector $\mathbf{y} \in \mathbb{R}^n$. However, it is known apriori that a certain number $k$ of these responses cannot be trusted since they are corrupted. These may correspond to corrupted pixels in visual recognition tasks or untrustworthy measurements in general sensing tasks.

Using these corrupted data points in any standard least-squares solver, especially when $k = \mathcal{O}(n)$, is likely to yield a poor model with little predictive power. A solution to this is to exclude corrupted

---

[*]Work done in part while Kush was a Research Fellow at Microsoft Research India.
[†]Work done in part while Kamalaruban was interning at Microsoft Research India.

Table 1: A comparison of different RLSR algorithms and their properties. CRR is the first efficient RLSR algorithm to guarantee consistency in the presence of a constant fraction of corruptions.

| Paper | Breakdown Point | Adversary | Consistent | Technique |
|-------|-----------------|-----------|------------|-----------|
| Wright & Ma, 2010 [21] | $\alpha \to 1$ | Oblivious | No | $L_1$ regularization |
| Chen & Dalalyan, 2010 [7] | $\alpha \geq \Omega(1)$ | Adaptive | No | SOCP |
| Chen et al., 2013 [6] | $\alpha \geq \Omega\left(\frac{1}{\sqrt{d}}\right)$ | Adaptive | No | Robust thresholding |
| Nguyen & Tran, 2013 [16] | $\alpha \to 1$ | Oblivious | No | $L_1$ regularization |
| Nguyen & Tran, 2013b [17] | $\alpha \to 1$ | Oblivious | No | $L_1$ regularization |
| McWilliams et al., 2014 [14] | $\alpha \geq \Omega\left(\frac{1}{\sqrt{d}}\right)$ | Oblivious | No | Weighted subsampling |
| Bhatia et al., 2015 [3] | $\alpha \geq \Omega(1)$ | Adaptive | No | Hard thresholding |
| **This paper** | $\alpha \geq \Omega(1)$ | **Oblivious** | **Yes** | **Hard thresholding** |

points from consideration. The RLSR problem formalizes this requirement as follows:

$$(\widehat{\mathbf{w}}, \widehat{S}) = \underset{\substack{\mathbf{w} \in \mathbb{R}^p, S \subset [n] \\ |S| = n-k}}{\arg\min} \sum_{i \in S} (y_i - \mathbf{x}_i^T \mathbf{w})^2, \tag{1}$$

This formulation seeks to simultaneously extract the set of uncorrupted points and estimate the least-squares solutions over those uncorrupted points. Due to the combinatorial nature of the RLSR formulation (1), solving it directly is challenging and in fact, NP-hard in general [3, 20].

Literature in robust statistics suggests several techniques to solve (1). The most common model assumes a realizable setting wherein there exists a gold model $\mathbf{w}^*$ that generates the non-corrupted responses. A vector of *corruptions* is then introduced to model the corrupted responses i.e.

$$\mathbf{y} = X^T \mathbf{w}^* + \mathbf{b}^*. \tag{2}$$

The goal of RLSR is to recover $\mathbf{w}^* \in \mathbb{R}^d$, the *true* model. The vector $\mathbf{b}^* \in \mathbb{R}^n$ is a $k$-sparse vector which takes non-zero values on at most $k$ corrupted samples out of the $n$ total samples, and a zero value elsewhere. A more useful, but challenging model is one in which (mostly heteroscedastic and i.i.d.) Gaussian noise is injected into the responses in addition to the corruptions.

$$\mathbf{y} = X^T \mathbf{w}^* + \mathbf{b}^* + \boldsymbol{\epsilon}. \tag{3}$$

Note that the Gaussian noise vector $\boldsymbol{\epsilon}$ is not sparse. In fact, we have $\|\boldsymbol{\epsilon}\|_0 = n$ almost surely.

## 2 Related Works

A string of recent works have looked at the RLSR problem in various settings. To facilitate a comparison among these, we set the following benchmarks for RLSR algorithms

1. (Breakdown Point) This is the number of corruptions $k$ that an RLSR algorithm can tolerate is a direct measure of its robustness. This limit is formalized as the *breakdown point* of the algorithm in statistics literature. The breakdown point $k$ is frequently represented as a fraction $\alpha$ of the total number of data points i.e. $k = \alpha \cdot n$.

2. (Adversary Model) RLSR algorithms frequently resort to an adversary model to specify how are the corruptions introduced into the regression problem. The strictest is the *adaptive adversarial* model wherein the adversary is able to view $X$ and $\mathbf{w}^*$ (as well as $\boldsymbol{\epsilon}$ if Gaussian noise is present) before deciding upon $\mathbf{b}^*$. A weaker model is the *oblivious adversarial* model wherein the adversary generates a $k$-sparse vector in complete ignorance of $X$ and $\mathbf{w}^*$ (and $\boldsymbol{\epsilon}$). However, the adversary is still free to make arbitrary choices for the location and values of corruptions.

3. (Consistency) RLSR algorithms that are able to operate in the *hybrid* noise model with sparse adversarial corruptions as well as dense Gaussian noise are more valuable. An RLSR algorithms is said to be consistent if, when invoked in the hybrid noise model on $n$ data points sampled from a distribution with appropriate characteristics, the RLSR algorithm returns an estimate $\widehat{\mathbf{w}}_n$ such that $\lim_{n \to \infty} \mathbb{E}\left[\widehat{\mathbf{w}}_n - \mathbf{w}^*\right]_2 = 0$ (for simplicity, assume a fixed covariate design with the expectation being over random Gaussian noise in the responses).

In Table 1, we present a summarized view of existing RLSR techniques and their performance vis-a-vis the benchmarks discussed above. Past work has seen the application of a wide variety of algorithmic techniques to solve this problem, including more expensive methods involving $L_1$ regularization (for example $\min_{\mathbf{w},\mathbf{b}} \lambda_w \|\mathbf{w}\|_1 + \lambda_b \|\mathbf{b}\|_1 + \|X^\top \mathbf{w} + \mathbf{b} - \mathbf{y}\|_2^2$) and second-order cone programs such as [21, 7, 16, 17], as well as more scalable methods such as the robust thresholding and iterative hard thresholding [6, 3]. As the work of [3] shows, $L_1$ regularization and other expensive methods struggle to scale to even moderately sized problems.

The adversary models considered by these works is also quite diverse. Half of the works consider an oblivious adversary and the other half brace themselves against an adaptive adversary. The oblivious adversary model, although weaker, can model some important practical situations where there is systematic error in the sensing equipment being used, such as a few pixels in a camera becoming unresponsive. Such errors are surely not random, and hence cannot be modeled as Gaussian noise, but introduce corruptions the final measurement in a manner that is oblivious of the signal actually being sensed, in this case the image being photographed.

An important point of consideration is the breakdown point of these methods. Among those cited in Table 1, the works of [21] and [16] obtain the best breakdown points that allow a fraction of points to be corrupted that is arbitrarily close to 1. They require the data to be generated from either an isotropic Gaussian ensemble or be row-sampled from an incoherent orthogonal matrix. Most results mentioned in the table allow a constant fraction of points to be corrupted i.e. allow $k = \alpha \cdot n$ corruptions for some fixed constant $\alpha > 0$. This is still impressive since it allows a dense subset of data points to be corrupted and yet guarantees recovery. However, as we shall see below, these results cannot guarantee consistency while allowing $k = \alpha \cdot n$ corruptions.

We note that we use the term *dense* to refer to the corruptions in our model since they are a constant fraction of the total available data. Moreover, as we shall see, this constant shall be universal and independent of the ambient dimensionality $d$. This terminology is used to contrast against some other works which can tolerate only $o(n)$ corruptions which is arguably much sparser. For instance, as we shall see below, the work of [17] can tolerate only $o(n/\log n)$ corruptions if a consistent estimate is expected. The work of [6] also offers a weak guarantee wherein they are only able to tolerate a $1/\sqrt{d}$ fraction of corruptions. However, [6] allow corruptions in covariates as well.

However, we note that *none* of the algorithms listed here, and to the best of our knowledge elsewhere as well, are able to guarantee a consistent solution, irrespective of assumptions on the adversary model. More specifically, none of these methods are able to guarantee exact recovery of $\mathbf{w}^*$, even with $n \to \infty$ and constant fraction of corruptions $\alpha = \Omega(1)$ (i.e. $k = \Omega(n)$). At best, they guarantee $\|\mathbf{w} - \mathbf{w}^*\|_2 \leq \mathcal{O}(\sigma)$ when $k = \Omega(n)$ where $\sigma$ is the standard deviation of the white noise (see Equation 3). Thus, their estimation error is of the order of the white noise in the system, even if the algorithm is supplied with an infinite amount of data. This is quite unsatisfactory, given our deep understanding of the consistency guarantees for least squares models.

For example, consider the work of [17] which considers a corruption model similar to (3). The work makes deterministic assumptions on the data matrix and proposes the following convex program.

$$\min_{\mathbf{w},\mathbf{b}} \lambda_w \|\mathbf{w}\|_1 + \lambda_b \|\mathbf{b}\|_1 + \|X^\top \mathbf{w} + \mathbf{b} - \mathbf{y}\|_2^2. \tag{4}$$

For Gaussian designs, which we also consider, their results guarantee that for $n = \mathcal{O}(s \log d)$,

$$\|\widehat{\mathbf{w}} - \mathbf{w}^*\|_2 + \|\widehat{\mathbf{b}} - \mathbf{b}^*\|_2 \leq \mathcal{O}\left(\sqrt{\frac{\sigma^2 s \log d \log n}{n}} + \sqrt{\frac{\sigma^2 k \log n}{n}}\right)$$

where $s$ is the sparsity index of the regressor $\mathbf{w}^*$. Note that for $k = \Theta(n)$, the right hand side behaves as $\Omega(\sigma \sqrt{\log n})$. Thus, the result is unable to ensure $\lim_{n \to \infty} \mathbb{E}[\widehat{\mathbf{w}}_n - \mathbf{w}^*]_2 = 0$.

We have excluded some classical approaches to the RLSR problem from the table such as [18, 1, 2] which use the Least Median of Squares (LMS) and Least Trimmed Squares (LTS) methods that guaranteed consistency but may require an exponential running time. Our focus is on polynomial time algorithms, more specifically those that are efficient and scalable. We note a recent work [5] in robust stochastic optimization which is able to tolerate a constant fraction of corruptions $\alpha \to 1$. However, their algorithms operate in the *list-decoding* model wherein they output not one, but as many as $\mathcal{O}\left(\frac{1}{1-\alpha}\right)$ models, of which one (unknown) model is guaranteed to be correct.

*Recovering Sparse High-dimensional Models*: We note that several previous works extend their methods and analyses to handle the case of sparse robust recovery in high-dimensional settings as well, including [3, 7, 17]. A benefit of such extensions is the ability to work even in data starved settings $n \ll d$ if the true model $\mathbf{w}^*$ is $s$-sparse with $s \ll d$. However, previous works continue to require the number of corruptions to be of the order of $k = o(n)$ or else $k = \mathcal{O}(n/s)$ in order to ensure that $\lim_{n \to \infty} \mathbb{E}[\widehat{\mathbf{w}}_n - \mathbf{w}^*]_2 = 0$ and cannot ensure consistency if $k = \mathcal{O}(n)$. This is evident, for example from the recovery guarantee offered by [17] discussed above, which requires $k = o(n/\log n)$. We do believe our CRR estimator can be adapted to high dimensional settings as well. However, the details are tedious and we reserve them for an expanded version of the paper.

## 3  Our Contributions

In this paper, we remedy the above problem by using a simple and scalable iterative hard-thresholding algorithm called CRR along with a novel two-stage proof technique. Given $n$ covariates that form a Gaussian ensemble, our method in time poly$(n, d)$, outputs an estimate $\widehat{\mathbf{w}}_n$ s.t. $\|\widehat{\mathbf{w}}_n - \mathbf{w}^*\|_2 \to 0$ as $n \to \infty$ (see Theorem 4 for a precise statement). In fact, our method guarantees a nearly optimal error rate of $\|\widehat{\mathbf{w}}_n - \mathbf{w}^*\|_2 \leq \sigma\sqrt{\frac{d}{n}}$. It is noteworthy that CRR can tolerate a constant fraction of corruptions i.e. tolerate $k = \alpha \cdot n$ corruptions for some fixed $\alpha > 0$.

We note that although hard thresholding techniques have been applied to the RLSR problem earlier [3, 6], none of those methods are able to guarantee a consistent solution to the problem. Our results hold in the setting where a *constant fraction* of the responses are corrupted by an *oblivious adversary* (i.e. the one which corrupts observations without information of the data points themselves). Our algorithm runs in time $\widetilde{\mathcal{O}}(d^3 + nd)$, where $d$ is the dimensionality of the data. Moreover, as we shall see, our technique makes more efficient use of data than previous hard thresholding methods such as TORRENT [3].

To the best of our knowledge, this is the *first* efficient and consistent estimator for the RLSR problem in the challenging setting where a *constant* fraction of the responses may be corrupted in the presence of dense noise. We would like to note that the problem of consistent robust regression is especially challenging because without the assumption of an *oblivious* adversary, consistent estimation with a constant fraction of corruptions (even for an arbitrarily small constant) may be impossible even when supplied with infinitely many data points.

However, by crucially using the restriction of obliviousness on the adversary along with a novel proof technique, we are able to provide a consistent estimator for RLSR with optimal (up to constants) statistical and computational complexity.

**Discussion on Problem Setting**: We clarify that our improvements come at a cost. Our results assume an oblivious adversary whereas several previous works allowed a fully adaptive adversary. Indeed there is no free-lunch: it seems unlikely that consistent estimators are even possible in the face of a fully adaptive adversary who can corrupt a constant fraction of responses since such an adversary can use his power to introduce biased noise into the model in order to defeat any estimator. An oblivious adversary is prohibited from looking at the responses before deciding the corruptions and is thus unable to do the above.

**Paper Organization**: We will begin our discussion by introducing the problem formulation, relevant notation, and tools in Section 4. This is followed by Section 5 where we develop CRR, a near-linear time algorithm that gives consistent estimates for the RLSR problem, which we analyze in Section 6. Finally in Section 7, we present rigorous experimental benchmarking of this algorithm. In Section 8 we offer some clarifications on how the manuscript was modified in response to reviewer comments.

## 4  Problem Formulation

We are given $n$ data points $X = [\mathbf{x}_1, \ldots, \mathbf{x}_n] \in \mathbb{R}^{d \times n}$, where $\mathbf{x}_i \in \mathbb{R}^d$ are the *covariates* and, for some *true* model $\mathbf{w}^* \in \mathbb{R}^d$, the vector of *responses* $\mathbf{y} \in \mathbb{R}^n$ is generated

$$\mathbf{y} = X^\top \mathbf{w}^* + \mathbf{b}^* + \boldsymbol{\epsilon}. \tag{5}$$

The responses suffer two kinds of perturbations – *dense white noise* $\epsilon_i \sim \mathcal{N}(0, \sigma^2)$ that is chosen in an i.i.d. fashion independently of the data $X$ and the model $\mathbf{w}^*$, and *adversarial corruptions*

---

**Algorithm 1** CRR: Consistent Robust Regression

---

**Input:** Covariates $X = [\mathbf{x}_1, \ldots, \mathbf{x}_n]$, responses $\mathbf{y} = [y_1, \ldots, y_n]^\top$, corruption index $k$, tolerance $\epsilon$
1: $\mathbf{b}^0 \leftarrow \mathbf{0}, t \leftarrow 0,$
   $P_X \leftarrow X^\top (XX^\top)^{-1} X$
2: **while** $\left\| \mathbf{b}^t - \mathbf{b}^{t-1} \right\|_2 > \epsilon$ **do**
3: $\quad \mathbf{b}^{t+1} \leftarrow \mathrm{HT}_k(P_X \mathbf{b}^t + (I - P_X)\mathbf{y})$
4: $\quad t \leftarrow t + 1$
5: **end while**
6: **return** $\mathbf{w}^t \leftarrow (XX^\top)^{-1} X(\mathbf{y} - \mathbf{b}^t)$

---

in the form of $\mathbf{b}^*$. We assume that $\mathbf{b}^*$ is a $k^*$-sparse vector albeit one with potentially unbounded entries. The constant $k^*$ will be called the *corruption index* of the problem. We assume the *oblivious adversary* model where $\mathbf{b}^*$ is chosen independently of $X, \mathbf{w}^*$ and $\boldsymbol{\epsilon}$.

Although there exist works that operate under a fully adaptive adversary [3, 7], none of these works are able to give a consistent estimate, whereas our algorithm CRR does provide a consistent estimate. We also note that existing works are unable to give consistent estimates even in the oblivious adversary model. Our result requires a significantly finer analysis; the standard $\ell_2$-norm style analysis used by existing works [3, 7] seems incapable of offering a consistent estimation result in the robust regression setting.

We will require the notions of *Subset Strong Convexity* and *Subset Strong Smoothness* similar to [3] and reproduce the same below. For any set $S \subset [n]$, let $X_S := [\mathbf{x}_i]_{i \in S} \in \mathbb{R}^{d \times |S|}$ denote the matrix with columns in that set. We define $\mathbf{v}_S$ for a vector $\mathbf{v} \in \mathbb{R}^n$ similarly. $\lambda_{\min}(X)$ and $\lambda_{\max}(X)$ will denote, respectively, the smallest and largest eigenvalues of a square symmetric matrix $X$.

**Definition 1** (SSC Property). *A matrix $X \in \mathbb{R}^{d \times n}$ is said to satisfy the* Subset Strong Convexity *Property at level $m$ with constant $\lambda_m$ if the following holds:*

$$\lambda_m \leq \min_{|S|=m} \lambda_{\min}(X_S X_S^\top)$$

**Definition 2** (SSS Property). *A matrix $X \in \mathbb{R}^{d \times n}$ is said to satisfy the* Subset Strong Smoothness *Property at level $m$ with constant $\Lambda_m$ if the following holds:*

$$\max_{|S|=m} \lambda_{\max}(X_S X_S^\top) \leq \Lambda_m.$$

Intuitively speaking, the SSC and SSS properties ensure that the regression problem remains well conditioned, even if restricted to an arbitrary subset of the data points. This allows the estimator to recover the exact model no matter what portion of the data was left uncorrupted by the adversary. We refer the reader to the Appendix A for SSC/SSS bounds for Gaussian ensembles.

## 5 CRR: A Hard Thresholding Approach to Consistent Robust Regression

We now present a consistent method CRR for the RLSR problem. CRR takes a significantly different approach to the problem than previous works. Instead of attempting to exclude data points deemed unclean (as done by the TORRENT algorithm proposed by [3]), CRR focuses on correcting the errors. This allows CRR to work with the entire dataset at all times, as opposed to TORRENT that works with a fraction of the data at any given point of time.

To motivate the CRR algorithm, we start with the RLSR formulation $\min_{\mathbf{w} \in \mathbb{R}^p, \|\mathbf{b}\|_0 \leq k^*} \frac{1}{2} \left\| X^\top \mathbf{w} - (\mathbf{y} - \mathbf{b}) \right\|_2^2$, and realize that given any estimate $\widehat{\mathbf{b}}$ of the corruption vector, the optimal model with respect to this estimate is given by the expression $\widehat{\mathbf{w}} = (XX^\top)^{-1} X(\mathbf{y} - \widehat{\mathbf{b}})$. Plugging this expression for $\widehat{\mathbf{w}}$ into the formulation allows us to reformulate the RLSR problem.

$$\min_{\|\mathbf{b}\|_0 \leq k^*} f(\mathbf{b}) = \frac{1}{2} \left\| (I - P_X)(\mathbf{y} - \mathbf{b}) \right\|_2^2 \tag{6}$$

where $P_X = X^\top (XX^\top)^{-1} X$. This greatly simplifies the problem by casting it as a *sparse parameter estimation* problem instead of a data subset selection problem (as done by TORRENT). CRR directly

optimizes (6) by using a form of iterative hard thresholding. Notice that this approach allows CRR to keep using the entire set of data points at all times, all the while using the current estimate of the parameter $\mathbf{b}$ to correct the errors in the observations. At each step, CRR performs the following update: $\mathbf{b}^{t+1} = \mathrm{HT}_k(\mathbf{b}^t - \nabla f(\mathbf{b}^t))$, where $k$ is a parameter for CRR. Any value $k \geq 2k^*$ suffices to ensure convergence and consistency, as will be clarified in the theoretical analysis. The hard thresholding operator $\mathrm{HT}_k(\cdot)$ is defined below.

**Definition 3** (Hard Thresholding). *For any $\mathbf{v} \in \mathbb{R}^n$, let the permutation $\sigma_{\mathbf{v}} \in S_n$ order elements of $\mathbf{v}$ in descending order of their magnitudes. Then for any $k \leq n$, we define the hard thresholding operator as $\widehat{\mathbf{v}} = HT_k(\mathbf{v})$ where $\widehat{\mathbf{v}}_i = \mathbf{v}_i$ if $\sigma_{\mathbf{v}}^{-1}(i) \leq k$ and 0 otherwise.*

We note that CRR functions with a fixed, unit step length, which is convenient in practice as it avoids step length tuning, something most IHT algorithms [12, 13] require. For simplicity of exposition, we will consider only Gaussian ensembles for the RLSR problem i.e. $\mathbf{x}_i \sim \mathcal{N}(\mathbf{0}, \Sigma)$; our proof technique works for general sub-Gaussian ensembles with appropriate distribution dependent parameters. Since CRR interacts with the data only using the projection matrix $P_X$, for Gaussian ensembles, one can assume without loss of generality that the data points are generated from a spherical Gaussian i.e. $\mathbf{x}_i \sim \mathcal{N}(\mathbf{0}, I_{d \times d})$. Our analysis will take care of the condition number of the data ensemble whenever it is apparent in the convergence rates.

Before moving to present the consistency and convergence guarantees for CRR, we note that Gaussian ensembles are known to satisfy the SSC/SSS properties with high probability. For instance, in the case of the standard Gaussian ensemble, we have SSC/SSS constants of the order of $\Lambda_m \leq \mathcal{O}\left(m\sqrt{\log \frac{n}{m}} + \sqrt{n}\right)$ and $\lambda_m \geq n - \mathcal{O}\left((n-m)\sqrt{\log \frac{n}{n-m}} + \sqrt{n}\right)$. These results are known from previous works [3, 10] and are reproduced in Appendix A.

## 6  Consistency Guarantees for CRR

**Theorem 4.** *Let $x_i \in \mathbb{R}^d, 1 \leq i \leq n$ be generated i.i.d. from a Gaussian distribution, let $y_i$'s be generated using (5) for a fixed $\mathbf{w}^*$, and let $\sigma^2$ be the noise variance. Also let the number of corruptions $k^*$ be s.t. $2k^* \leq k \leq n/10000$. Then for any $\epsilon, \delta > 0$, with probability at least $1 - \delta$, after $\mathcal{O}\left(\log \frac{\|\mathbf{b}^*\|_2}{\sigma k + \epsilon} + \log \frac{n}{d}\right)$ steps, CRR ensures that $\|\mathbf{w}^t - \mathbf{w}^*\|_2 \leq \epsilon + \mathcal{O}\left(\frac{\sigma}{\sqrt{\lambda_{\min}(\Sigma)}}\sqrt{\frac{d}{n}\log \frac{nd}{\delta}}\right)$.*

The above result establishes consistency of the CRR method with an error rate of $\tilde{\mathcal{O}}(\sigma\sqrt{d/n})$ that is known to be statistically optimal. It is notable that this optimal rate is being ensured in the presence of gross and unbounded outliers. We reiterate that to the best of our knowledge, this is the first instance of a poly-time algorithm being shown to be consistent for the RLSR problem. It is also notable that the result allows the corruption index to be $k^* = \Omega(n)$, i.e. allows upto a *constant* factor of the total number of data points to be arbitrarily corrupted, while ensuring consistency, which existing results [3, 6, 16] do not ensure.

We pause a bit to clarify some points regarding the result. Firstly we note that the upper bound on recovery error consists of two terms. The first term is $\epsilon$ which can be made arbitrarily small simply by executing the CRR algorithm for several iterations. The second term is more crucial and underscores the consistency properties of CRR. The second term is of the form $\mathcal{O}\left(\sigma\sqrt{d\log(nd)/n}\right)$ and is easily seen to vanish with $n \to \infty$ for any constant $d, \sigma$. Secondly we note that the result requires $k^* \leq n/20000$ i.e. $\alpha \leq 1/20000$. Although this constant might seem small, we stress that these constants are not the best possible since we preferred analyses that were more accessible. Indeed, in our experiments, we found CRR to be robust to much higher corruption levels than what the Theorem 4 guarantees. Thirdly, we notice that the result requires the CRR to be executed with the corruption index set to a value $k \geq 2k^*$. In practice the value of $k$ can be easily tuned using a simple binary search because of the speed of execution that CRR offers (see Section 7).

For our analysis, we will divide CRR's execution into two phases – a *coarse convergence* phase and a *fine convergence* phase. CRR will enjoy a linear rate of convergence in both phases. However, the coarse convergence analysis will only ensure $\|\mathbf{w}^t - \mathbf{w}^*\|_2 = \mathcal{O}(\sigma)$. The fine convergence phase will then use a much more careful analysis of the algorithm to show that in at most $\mathcal{O}(\log n)$ more

iterations, CRR ensures $\|\mathbf{w}^t - \mathbf{w}^*\|_2 = \tilde{\mathcal{O}}(\sigma\sqrt{d/n})$, thus establishing consistency of the method. Existing methods, such as TORRENT, ensure an error level of $\mathcal{O}(\sigma)$, but no better.

As shorthand notation, let $\boldsymbol{\lambda}^t := (XX^\top)^{-1}X(\mathbf{b}^t - \mathbf{b}^*)$, $\mathbf{g} := (I - P_X)\boldsymbol{\epsilon}$, and $\mathbf{v}^t = X^\top \boldsymbol{\lambda}^t + \mathbf{g}$. Let $S^* := \text{supp}(\mathbf{b}^*)$ be the true locations of the corruptions and $I^t := \text{supp}(\mathbf{b}^t) \cup \text{supp}(\mathbf{b}^*)$.

**Coarse convergence**: Here we establish a result that guarantees that after a certain number of steps $T_0$, CRR identifies the corruption vector with a relatively high accuracy and consequently ensures that $\|\mathbf{w}^{T_0} - \mathbf{w}^*\|_2 \leq \mathcal{O}(\sigma)$.

**Lemma 5.** *For any data matrix $X$ that satisfies the SSC and SSS properties such that $\frac{2\Lambda_{k+k^*}}{\lambda_n} < 1$, CRR, when executed with $k \geq k^*$, ensures for any $\epsilon, \delta > 0$, with probability at least $1 - \delta$ (over the random Gaussian noise $\epsilon$ in the responses – see (3)) that after $T_0 = \mathcal{O}\left(\log \frac{\|\mathbf{b}^*\|_2}{e_0 + \epsilon}\right)$ steps, $\|\mathbf{b}^{T_0} - \mathbf{b}^*\|_2 \leq 3e_0 + \epsilon$, where $e_0 = \mathcal{O}\left(\sigma\sqrt{(k + k^*)\log\frac{n}{\delta(k+k^*)}}\right)$ for standard Gaussian designs.*

Using Lemma 12 (see the appendix), we can translate the above result to show that $\|\mathbf{w}^{T_0} - \mathbf{w}^*\|_2 \leq 0.95\sigma + \epsilon$, assuming $k^* \leq k \leq \frac{n}{150}$. However, Lemma 5 will be more useful in the following fine convergence analysis.

**Fine convergence**: We now show that CRR progresses further at a linear rate to achieve a consistent solution. In Lemma 6, we show that $\|X(\mathbf{b}^t - \mathbf{b}^*)\|_2$ has a linear decrease for every iteration $t > T_0$ along with a term which is $\tilde{\mathcal{O}}(\sqrt{dn})$. The proof proceeds by showing that for any fixed $\boldsymbol{\lambda}^t$ such that $\|\boldsymbol{\lambda}^t\|_2 \leq \frac{\sigma}{100}$, we obtain a linear decrease in $\|\boldsymbol{\lambda}^{t+1}\|_2 = \|(XX^T)^{-1}X(\mathbf{b}^{t+1} - \mathbf{b}^*)\|_2$. We then take a union bound over a fine $\epsilon$-net over all possible values of $\boldsymbol{\lambda}^t$ to obtain the final result.

**Lemma 6.** *Let $X = [\mathbf{x}_1, \mathbf{x}_2, \ldots, \mathbf{x}_n]$ be a data matrix consisting of i.i.d. standard normal vectors i.e $\mathbf{x}_i \sim \mathcal{N}(\mathbf{0}, I_{d\times d})$, and $\boldsymbol{\epsilon} \sim \mathcal{N}(0, \sigma^2 \cdot I_{n\times n})$ be a standard normal vector of white noise values drawn independently of $X$. For any $\boldsymbol{\lambda} \in \mathbb{R}^d$ such that $\|\boldsymbol{\lambda}\|_2 \leq \frac{\sigma}{100}$, define $\mathbf{b}^{new} = HT_k(X^\top\boldsymbol{\lambda} + \boldsymbol{\epsilon} + \mathbf{b}^*)$, $\mathbf{z}^{new} = \mathbf{b}^{new} - \mathbf{b}^*$ and $\boldsymbol{\lambda}^{new} = (XX^T)^{-1}X\mathbf{z}^{new}$, where $k \geq 2k^*$, $|supp(\mathbf{b}^*)| \leq k^*$, $k^* \leq n/10000$, and $d \leq n/10000$. Then, with probability at least $1 - 1/n^5$, for every $\boldsymbol{\lambda}$ s.t. $\|\boldsymbol{\lambda}\|_2 \leq \frac{\sigma}{100}$, we have*

$$\|X\mathbf{z}^{new}\|_2 \leq .9n\|\boldsymbol{\lambda}\|_2 + 100\sigma\sqrt{d \cdot n}\log^2 n,$$

$$\|\boldsymbol{\lambda}^{new}\|_2 \leq .91\|\boldsymbol{\lambda}\|_2 + 110\sigma\sqrt{\frac{d}{n}}\log^2 n.$$

Putting all these results together establishes Theorem 4. See Appendix B for a detailed proof. Note that while both the coarse/fine stages offer a linear rate of convergence, it is the fine phase that ensures consistency. Indeed, the coarse phase only acts as a sort of good-enough initialization. Several results in non-convex optimization assume a nice initialization "close" to the optimum (alternating minimization, EM etc). In our case, we have a happy situation where the initialization and main algorithms are one and the same. Note that we could have actually used other algorithms e.g. TORRENT to perform initialization as well since TORRENT [3, Theorem 10] essentially offers the same (weak) guarantee as Lemma 5 offers.

## 7 Experiments

Experiments were carried out on synthetically generated linear regression datasets with corruptions. All implementations were done in Matlab and were run on a single core 2.4GHz machine with 8GB RAM. The experiments establish the following: 1) CRR gives consistent estimates of the regression model, especially in situations with a large number of corruptions where the ordinary least squares estimator fails catastrophically, 2) CRR scales better to large datasets than the TORRENT-FC algorithm of [3] (upto $5\times$ faster) and the Extended Lasso algorithm of [17] (upto $20\times$ faster). The main reason behind this speedup is that TORRENT keeps changing its mind on which active set of points it wishes to work with. Consequently, it expends a lot of effort processing each active set. CRR on the other hand does not face such issues since it always works with the entire set of points.

**Data**: The model $\mathbf{w}^* \in \mathbb{R}^d$ was chosen to be a random unit norm vector. The data was generated as $\mathbf{x}_i \sim \mathcal{N}(0, I_d)$. The $k^*$ responses to be corrupted were chosen uniformly at random and the

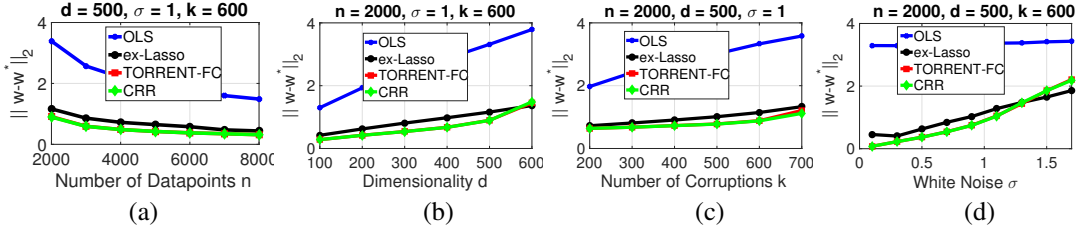

Figure 1: Variation of recovery error with varying number of data points $n$, dimensionality $d$, number of corruptions $k^*$ and white noise variance $\sigma$. CRR and TORRENT show better recovery properties than the non-robust OLS on all experiments. Extended Lasso offers comparable or slightly worse recovery in most settings. Figure 1(a) ascertains the $\widetilde{\mathcal{O}}\left(\sqrt{1/n}\right)$-consistency of CRR as is shown in the theoretical analysis.

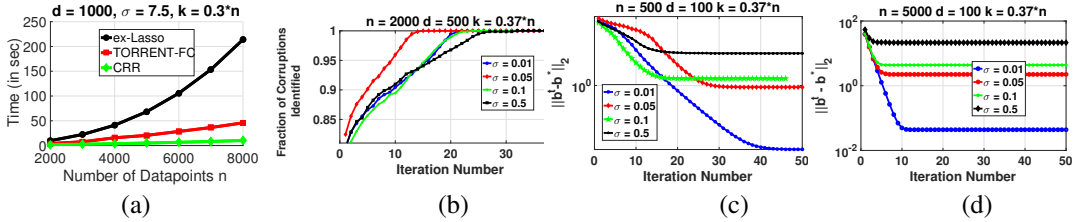

Figure 2: Figure 2(a) show the average CPU run times of CRR, TORRENT and Extended Lasso with varying sample sizes. CRR can be an order of magnitude faster than TORRENT and Extended Lasso on problems in 1000 dimensions while ensuring similar recovery properties.. Figure 2(b), 2(c) and 2(d) show that CRR eventually not only captures the total mass of corruptions, but also does support recovery of the corrupted points in an accurate manner. With every iteration, CRR improves upon its estimate of $\mathbf{b}^*$ and provides cleaner points for estimation of $\mathbf{w}$. CRR is also able to very effectively utilize larger data sets to offer much faster convergence. Notice the visibly faster convergence in Figure 2(d) which uses 10x more points than figure (c).

value of the corruptions was sets as $b_i^* \sim \mathrm{Unif}\,(10, 20)$. Responses were then generated as $y_i = \langle \mathbf{x}_i, \mathbf{w}^* \rangle + \eta_i + b_i^*$ where $\eta_i \sim \mathcal{N}(0, \sigma^2)$. All reported results were averaged over 20 randomly trials.

**Evaluation Metric**: We measure the performance of various algorithms using the standard $L_2$ error: $r_{\widehat{\mathbf{w}}} = \|\widehat{\mathbf{w}} - \mathbf{w}^*\|_2$. For the timing experiments, we deemed an algorithm to converge on an instance if it obtained a model $\mathbf{w}^t$ such that $\|\mathbf{w}^t - \mathbf{w}^{t-1}\|_2 \leq 10^{-4}$.

**Baseline Algorithms**: CRR was compared to two baselines 1) the Ordinary Least Squares (OLS) estimator which is oblivious of the presence of any corruptions in the responses, 2) the TORRENT algorithm of [3] which is a recently proposed method for performing robust least squares regression, and 3) the Extended Lasso (ex-Lasso) approach of [15] for which we use the FISTA implementation of [23] and choose the regularization paramaters for our model data as mentioned by the authors.

**Recovery Properties & Timing**: CRR, TORRENT and ex-Lasso were found to be competitive, and offered much lower residual errors $\|\mathbf{w} - \mathbf{w}^*\|_2$ than the non-robust OLS method when varying dataset size Figure 1(a), dimensionality Figure 1(b), number of corrupted responses Figure 1(c), and magnitude of white noise Figure 1(d). In terms of scaling properties, CRR exhibited faster runtimes than TORRENT-FC as depicted in Figure 2(a). CRR can be upto $5\times$ faster than TORRENT and upto $20\times$ faster than ex-Lasso on problems of 1000 dimensions. Figure 2(a) suggests that executing both TORRENT and ex-Lasso becomes very expensive with an order of magnitude increase in the dimension parameter of the problem while CRR scales gracefully. Also, Figures 2(c) and 2(d) show the variation of $\|\mathbf{b}^t - \mathbf{b}^*\|_2$ for various values of the noise parameter $\sigma$. The plot depicts the fact that as $\sigma \to 0$, CRR is correctly able to identify all the corrupted points and estimate the level of corruption correctly, thereby returning the exact solution $\mathbf{w}^*$. Notice that in Figure 2(d) which utilizes more data points, CRR offers uniformly faster convergence across all white noise levels.

**Choice of Potential Function**: In Lemmata 5 and 6, we show that $\|\mathbf{b}^t - \mathbf{b}^*\|_2$ decreases with every iteration. Figures 2(c) and (d) back this theoretical statement by showing that CRR's estimate of $\mathbf{b}^*$ improves with every iteration. Along with estimating the magnitude of $\mathbf{b}^*$, Figure 2(b) shows that CRR is also able to correctly identify the support of the corrupted points with increasing iterations.

## 8  Response to Reviewer Comments

We are thankful to the reviewers for their comments aimed at improving the manuscript. Below we offer some clarifications regarding the same.

1. We have fixed all typographical errors pointed out in the reviews.

2. We have included additional references as pointed out in the reviews.

3. We have improved the presentation of the statement of the results to make the theorem and lemma statements more crisp and self contained.

4. We have fixed minor inconsistencies in the figures by executing experiments afresh.

5. We note that CRR's reduction of the robust recovery problem to sparse recovery is not only novel, but also one that offers impressive speedups in practice over the fully corrective version of the existing TORRENT algorithm [3]. However, note that the reduction to sparse recovery actually hides a sort of "fully-corrective" step wherein the optimal model for a particular corruption estimate is used internally in the formulation. Thus, CRR is implicitly a fully corrective algorithm as well.

6. We agree with the reviewers that further efforts are needed to achieve results with sharper constants. For example, CRR offers robustness upto a breakdown fraction of 1/20000 which, although a constant, nevertheless leaves room for improvement. Having shown for the first time that tolerating a non-trivial, universally constant fraction of corruptions is possible in polynomial time, it is indeed encouraging to study how far can the breakdown point be pushed for various families of algorithms.

7. Our current efforts are aimed at solving the robust sparse recovery problems in high dimensional settings in a statistically consistent manner, as well as extending the consistency properties established in this paper for non-Gaussian, for example fixed, designs.

## Acknowledgments

The authors thank the reviewers for useful comments. PKar is supported by the Deep Singh and Daljeet Kaur Faculty Fellowship and the Research-I Foundation at IIT Kanpur, and thanks Microsoft Research India and Tower Research for research grants. KB gratefully acknowledges the support of the NSF through grant IIS-1619362.

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
