[Supplementary Material]

# Supplementary Material for Consistent Robust Regression

## A    SSC/SSS guarantees

In this section we restate some results from [3] which are required for the convergence analysis of the RLSR problem. Similar variants are known from other works, e.g. [10], as well.

**Definition 7.** *A random variable $x \in \mathbb{R}$ is called sub-Gaussian if the following quantity is finite*

$$\sup_{p \geq 1} p^{-1/2} \left( \mathbb{E}\left[|x|^p\right] \right)^{1/p}.$$

*Moreover, the smallest upper bound on this quantity is referred to as the sub-Gaussian norm of $x$ and denoted as $\|x\|_{\psi_2}$.*

**Definition 8.** *A vector-valued random variable $\mathbf{x} \in \mathbb{R}^p$ is called sub-Gaussian if its unidimensional marginals $\langle \mathbf{x}, \mathbf{v} \rangle$ are sub-Gaussian for all $\mathbf{v} \in S^{p-1}$. Moreover, its sub-Gaussian norm is defined as follows*

$$\|X\|_{\psi_2} := \sup_{\mathbf{v} \in S^{p-1}} \|\langle \mathbf{x}, \mathbf{v} \rangle\|_{\psi_2}$$

**Lemma 9.** *Let $X \in \mathbb{R}^{p \times n}$ be a matrix whose columns are sampled i.i.d from a standard Gaussian distribution i.e. $\mathbf{x}_i \sim \mathcal{N}(0, I)$. Then for any $\epsilon > 0$, with probability at least $1 - \delta$, $X$ satisfies*

$$\lambda_{\max}(XX^\top) \leq n + (1 - 2\epsilon)^{-1} \sqrt{cnp + c'n \log \frac{2}{\delta}}$$

$$\lambda_{\min}(XX^\top) \geq n - (1 - 2\epsilon)^{-1} \sqrt{cnp + c'n \log \frac{2}{\delta}},$$

*where $c = 24e^2 \log \frac{3}{\epsilon}$ and $c' = 24e^2$.*

**Theorem 10.** *Let $X \in \mathbb{R}^{p \times n}$ be a matrix whose columns are sampled i.i.d from a standard Gaussian distribution i.e. $\mathbf{x}_i \sim \mathcal{N}(0, I)$. Then for any $k > 0$, with probability at least $1 - \delta$, the matrix $X$ satisfies the SSC and SSS properties with constants*

$$\Lambda_k \leq k \left( 1 + 3e \sqrt{6 \log \frac{en}{k}} \right) + \mathcal{O}\left( \sqrt{np + n \log \frac{1}{\delta}} \right)$$

$$\lambda_k \geq n - (n - k) \left( 1 + 3e \sqrt{6 \log \frac{en}{n - k}} \right) - \Omega\left( \sqrt{np + n \log \frac{1}{\delta}} \right).$$

**Lemma 11.** *Let $X \in \mathbb{R}^{p \times n}$ be a matrix with columns sampled from some sub-Gaussian distribution with sub-Gaussian norm $K$ and covariance $\Sigma$. Then, for any $\delta > 0$, with probability at least $1 - \delta$, each of the following statements holds true:*

$$\lambda_{\max}(XX^\top) \leq \lambda_{\max}(\Sigma) \cdot n + C_K \cdot \sqrt{pn} + t\sqrt{n}$$

$$\lambda_{\min}(XX^\top) \geq \lambda_{\min}(\Sigma) \cdot n - C_K \cdot \sqrt{pn} - t\sqrt{n},$$

*where $t = \sqrt{\frac{1}{c_K} \log \frac{2}{\delta}}$, and $c_K, C_K$ are absolute constants that depend only on the sub-Gaussian norm $K$ of the distribution.*

## B    Convergence Proofs for CRR

**Theorem 4.** *Let $x_i \in \mathbb{R}^d, 1 \leq i \leq n$ be generated i.i.d. from a Gaussian distribution, let $y_i$'s be generated using (5) for a fixed $\mathbf{w}^*$, and let $\sigma^2$ be the noise variance. Also let the number of corruptions $k^*$ be s.t. $2k^* \leq k \leq n/10000$. Then for any $\epsilon, \delta > 0$, with probability at least $1 - \delta$, after $\mathcal{O}\left( \log \frac{\|\mathbf{b}^*\|_2}{\sigma k + \epsilon} + \log \frac{n}{d} \right)$ steps, CRR ensures that $\|\mathbf{w}^t - \mathbf{w}^*\|_2 \leq \epsilon + \mathcal{O}\left( \frac{\sigma}{\sqrt{\lambda_{\min}(\Sigma)}} \sqrt{\frac{d}{n} \log \frac{nd}{\delta}} \right)$.*

*Proof.* Using Lemma 5, after $\mathcal{O}\left( \log \frac{\|\mathbf{b}^*\|_2}{\sigma k + \epsilon} \right)$ steps, we get $\mathbf{b}^t$ s.t. $\|\boldsymbol{\lambda}^t\| = \|(X^T X)^{-1} X(\mathbf{b}^t - \mathbf{b}^*)\| \leq \frac{90(k^* + \sqrt{nd})}{n} \|\mathbf{b}^t - \mathbf{b}^*\| \leq \frac{90(k^* + \sqrt{nd})}{n} (6\sigma\sqrt{k^* \log n}) \leq \sigma/100$ as long as $k \leq n/10000$ and $n \geq d/10000$.

Now, recall that $\mathbf{b}_{t+1} = HT_k(\mathbf{b}^* + X^T(\boldsymbol{\lambda}^t - P_X \boldsymbol{\epsilon}) + \boldsymbol{\epsilon})$. Now, using Lemma 13 with $\boldsymbol{\lambda} = \boldsymbol{\lambda}^t - P_X \boldsymbol{\epsilon}$, we get:

$$\|\boldsymbol{\lambda}^{t+1}\|_2 \leq 0.91 \|\boldsymbol{\lambda}^t\|_2 + 110\sigma \sqrt{\frac{d}{n}} \log^2 n,$$

which ensures a linear convergence of the terms $\|\boldsymbol{\lambda}^t\|_2$ to a value $\epsilon + \mathcal{O}\left( \sigma \sqrt{\frac{d}{n} \log \frac{nd}{\delta}} \right)$.    □

**Lemma 5.** *For any data matrix $X$ that satisfies the SSC and SSS properties such that $\frac{2\Lambda_{k+k^*}}{\lambda_n} < 1$, CRR, when executed with $k \geq k^*$, ensures for any $\epsilon, \delta > 0$, with probability at least $1 - \delta$ (over the random Gaussian noise $\epsilon$ in the responses – see (3)) that after $T_0 = \mathcal{O}\left(\log \frac{\|\mathbf{b}^*\|_2}{e_0 + \epsilon}\right)$ steps, $\left\|\mathbf{b}^{T_0} - \mathbf{b}^*\right\|_2 \leq 3e_0 + \epsilon$, where $e_0 = \mathcal{O}\left(\sigma\sqrt{(k + k^*)\log \frac{n}{\delta(k+k^*)}}\right)$ for standard Gaussian designs.*

*Proof.* We start with the update step in CRR, and use the fact that $\mathbf{y} = X^\top \mathbf{w}^* + \mathbf{b}^* + \epsilon$ to rewrite the update as

$$\mathbf{b}^{t+1} \leftarrow \mathrm{HT}_k(P_X \mathbf{b}^t + (I - P_X)(X^\top \mathbf{w}^* + \mathbf{b}^* + \epsilon)).$$

Since $X^\top = P_X X^\top$, we get, using the notation set up before,

$$\mathbf{b}^{t+1} \leftarrow \mathrm{HT}_k(\mathbf{b}^* + X^\top \boldsymbol{\lambda}^t + \mathbf{g}).$$

Since $k \geq k^*$, using the properties of the hard thresholding step gives us

$$\left\|\mathbf{b}^{t+1}_{I^{t+1}} - (\mathbf{b}^*_{I^{t+1}} + X^\top_{I^{t+1}}\boldsymbol{\lambda}^t + \mathbf{g}_{I^{t+1}})\right\|_2 \leq \left\|\mathbf{b}^*_{I^{t+1}} - (\mathbf{b}^*_{I^{t+1}} + X^\top_{I^{t+1}}\boldsymbol{\lambda}^t + \mathbf{g}_{I^{t+1}})\right\|_2 = \left\|X^\top_{I^{t+1}}\boldsymbol{\lambda}^t + \mathbf{g}_{I^{t+1}}\right\|_2.$$

This, upon applying the triangle inequality, gives us

$$\left\|\mathbf{b}^{t+1} - \mathbf{b}^*\right\|_2 \leq 2\left\|X^\top_{I^{t+1}}\boldsymbol{\lambda}^t + \mathbf{g}_{I^{t+1}}\right\|_2.$$

Now, using the SSC and SSS properties of $X$, we can show that $\left\|X^\top_{I^{t+1}}\boldsymbol{\lambda}^t\right\|_2 = \left\|X^\top_{I^{t+1}}(XX^\top)^{-1}X_{I^t}(\mathbf{b}^t - \mathbf{b}^*)\right\|_2 \leq \frac{\Lambda_{k+k^*}}{\lambda_n}\left\|\mathbf{b}^t - \mathbf{b}^*\right\|_2$.

Since $\epsilon$ is a Gaussian vector, using tail bounds for Chi-squared random variables (for example, see Lemma 20 in [3]), for any set $S$ of size $k + k^*$, we have with probability at least $1 - \delta$, $\|\epsilon_S\|_2^2 \leq \sigma^2(k + k^*) + 2e\sigma^2\sqrt{6(k+k^*)\log\frac{1}{\delta}}$. Taking a union bound over all sets of size $(k + k^*)$ and $\binom{n}{k} \leq \left(\frac{en}{k}\right)^k$ gives us, with probability at least $1 - \delta$, for all sets $S$ of size at most $(k + k^*)$,

$$\|\epsilon_S\|_2 \leq \sigma\sqrt{(k+k^*)}\sqrt{1 + 2e\sqrt{6\log\frac{en}{\delta(k+k^*)}}}$$

Using tail bounds on Gaussian random variables[3], we can also show that for every $i$, with probability at least $1 - \delta$, we have $\|(X\epsilon)_i\|_2 \leq \sigma\left\|(X^\top)_i\right\|_2\sqrt{2\log\frac{1}{\delta}}$. Taking a union bound gives us, with the same confidence, $\|X\epsilon\|_2^2 \leq 2\sigma^2\|X\|_F^2\log\frac{d}{\delta} \leq 2\sigma^2 d\Lambda_n\log\frac{d}{\delta}$. This allows us to bound $\|\mathbf{g}_{I^{t+1}}\|_2$

$$\|\mathbf{g}_{I^{t+1}}\|_2 = \left\|\epsilon_{I^{t+1}} - X^\top_{I^{t+1}}(XX^\top)^{-1}X\epsilon\right\|_2$$

$$\leq \sigma\sqrt{(k+k^*)}\sqrt{1 + 2e\sqrt{6\log\frac{en}{\delta(k+k^*)}}} + \sigma\frac{\sqrt{\Lambda_{k+k^*}\Lambda_n}}{\lambda_n}\sqrt{2d\log\frac{d}{\delta}}$$

$$\leq \underbrace{\sigma\sqrt{(k+k^*)}\sqrt{1 + 2e\sqrt{6\log\frac{en}{\delta(k+k^*)}}}\left(1 + \sqrt{\frac{2d}{n}\log\frac{d}{\delta}}\right)}_{e_0}$$

$$= 1.0003e_0,$$

where the second last step is true for Gaussian designs and sufficiently large enough $n$. Note that $e_0$ does not depend on the iterates and is thus, a constant. This gives us

$$\left\|\mathbf{b}^{t+1} - \mathbf{b}^*\right\|_2 \leq \frac{2\Lambda_{k+k^*}}{\lambda_n}\left\|\mathbf{b}^t - \mathbf{b}^*\right\|_2 + 2.0006e_0.$$

For data matrices sampled from Gaussian ensembles, whose SSC and SSS properties will be established later, assuming $n \geq d\log d$, we have $e_0 = \mathcal{O}\left(\sigma\sqrt{(k + k^*)\log\frac{n}{\delta(k+k^*)}}\right)$. Thus, if $\frac{2\Lambda_{k+k^*}}{\lambda_n} < 1$, then in $T_0 = \mathcal{O}\left(\log\frac{\|\mathbf{b}^*\|_2}{e_0 + \epsilon}\right)$ steps, CRR ensures that $\left\|\mathbf{b}^{T_0} - \mathbf{b}^*\right\|_2 \leq 2.0009e_0 + \epsilon$. $\qquad\square$

**Lemma 12.** *Let $\lambda_{\min}(\Sigma)$ be the smallest eigenvalue of the covariance matrix of the distribution $\mathcal{N}(\mathbf{0}, \Sigma)$ that generates the data points. Then at any time instant t, we have $\|\mathbf{w}^t - \mathbf{w}^*\|_2 \leq \frac{2}{\sqrt{\lambda_{\min}(\Sigma)}} \left( 2\sigma\sqrt{\frac{d}{n}\log\frac{d}{\delta}} + \|\boldsymbol{\lambda}^t\|_2 \right)$.*

*Proof.* As described in Algorithm 1, $\mathbf{w}^t = (XX^\top)^{-1}X(\mathbf{y} - \mathbf{b}^t) = \mathbf{w}^* + (XX^\top)^{-1}X(\boldsymbol{\epsilon} + \mathbf{b}^* - \mathbf{b}^t)$. If we let $\overline{X} = \Sigma^{-1/2}X$, we get:

$$
\begin{aligned}
\left\|\mathbf{w}^t - \mathbf{w}^*\right\|_2 &\leq \frac{1}{\sqrt{\lambda_{\min}(XX^\top)}} \left\|X^\top(\mathbf{w}^t - \mathbf{w}^*)\right\|_2 \\
&\leq \frac{1}{\sqrt{n\lambda_{\min}(\Sigma) - C_\Sigma\sqrt{n}}} \left\|X^\top(\mathbf{w}^t - \mathbf{w}^*)\right\|_2 \\
&= \frac{1}{\sqrt{n\lambda_{\min}(\Sigma) - C_\Sigma\sqrt{n}}} \left\|X^\top(XX^\top)^{-1}X(\boldsymbol{\epsilon} + \mathbf{b}^* - \mathbf{b}^t)\right\|_2 \\
&= \frac{1}{\sqrt{n\lambda_{\min}(\Sigma) - C_\Sigma\sqrt{n}}} \left\|\overline{X}^\top(\overline{X}\,\overline{X}^\top)^{-1}\overline{X}(\boldsymbol{\epsilon} + \mathbf{b}^* - \mathbf{b}^t)\right\|_2 \\
&\leq \frac{\sqrt{\Lambda_n}}{\sqrt{n\lambda_{\min}(\Sigma) - C_\Sigma\sqrt{n}}} \left\|(\overline{X}\,\overline{X}^\top)^{-1}\overline{X}(\boldsymbol{\epsilon} + \mathbf{b}^* - \mathbf{b}^t)\right\|_2 \\
&\leq \frac{2}{\sqrt{\lambda_{\min}(\Sigma)}} \left( 2\sigma\sqrt{\frac{d}{n}\log\frac{d}{\delta}} + \|\boldsymbol{\lambda}^t\|_2 \right),
\end{aligned}
$$

where the second step follows from results on eigenvalue bounds for data matrices drawn from non-spherical Gaussians, where $C_\Sigma$ is a constant dependent on the subGaussian norm of the distribution. The fourth step is a whitening step executed for sake of convenience alone and can be bypassed. The step uses the fact that even though $X$ may be sampled from a non-spherical Gaussian $\mathcal{N}(\mathbf{0}, \Sigma)$, the quantity $X^\top(XX^\top)^{-1}X$ is distributed as $\overline{X}^\top(\overline{X}\,\overline{X}^\top)^{-1}\overline{X}$ where $\overline{X}$ is sampled from a spherical Gaussian $\mathcal{N}(\mathbf{0}, I)$. The last step assumes $n \geq \frac{2C_\Sigma}{\lambda_{\min}(\Sigma)}$ and uses the proof technique used in Lemma 5 to get

$$
\left\|(\overline{X}\,\overline{X}^\top)^{-1}\overline{X}\boldsymbol{\epsilon}\right\|_2 \leq \sigma\frac{\sqrt{\Lambda_n}}{\lambda_n}\sqrt{2d\log\frac{d}{\delta}} \leq 2\sigma\sqrt{\frac{d}{n}\log\frac{d}{\delta}}. \qquad \square
$$

**Lemma 6.** *Let $X = [\mathbf{x}_1, \mathbf{x}_2, \ldots, \mathbf{x}_n]$ be a data matrix consisting of i.i.d. standard normal vectors i.e $\mathbf{x}_i \sim \mathcal{N}(\mathbf{0}, I_{d\times d})$, and $\boldsymbol{\epsilon} \sim N(0, \sigma^2 \cdot I_{n\times n})$ be standard normal vector drawn independently of $X$. For any $\boldsymbol{\lambda} \in \mathbb{R}^d$ such that $\|\boldsymbol{\lambda}\|_2 \leq \frac{\sigma}{100}$, define $\mathbf{z} = HT_k(X^\top\boldsymbol{\lambda} + \boldsymbol{\epsilon} + \mathbf{b}^*) - \mathbf{b}^*$, where $k = 2k^*$ and $|supp(\mathbf{b}^*)| \leq k^*$. Also let $k^* \leq n/10000$ and $d \leq n/10000$. Then, the following holds (w.p. $\geq 1 - 1/n^5$):*

$$
\|X\mathbf{z}\| \leq .9n\|\boldsymbol{\lambda}\| + 300\sigma\sqrt{d \cdot n}\log^2 n
$$

*Proof.* We first decompose $\|X\mathbf{z}\|_2^2$, using the Pythagorean theorem, as:

$$
\|X\mathbf{z}\|_2^2 \leq \frac{1}{\|\boldsymbol{\lambda}\|_2^2}(\boldsymbol{\lambda}^T X\mathbf{z})^2 + \max_{\mathbf{v}, \|\mathbf{v}\|_2=1, \mathbf{v}^T\boldsymbol{\lambda}=0}(\mathbf{v}^T X\mathbf{z})^2. \tag{7}
$$

We now consider the first term above. Let $\tau_k > 0$ be such that the $k$ largest elements (in magnitude) of $\mathbf{b} = X^\top\boldsymbol{\lambda} + \boldsymbol{\epsilon} + \mathbf{b}^*$ are all greater than $\tau_k$ and the $(n - k)$ smallest elements (in magnitude) of $\mathbf{b}$ are all less than $\tau_k$.

That is,

$$
\begin{aligned}
\boldsymbol{\lambda}^\top X\mathbf{z} &= \sum_j (\boldsymbol{\lambda}^\top\mathbf{x}_j)(\mathbb{I}\left\{|\mathbf{x}_j^\top\boldsymbol{\lambda} + b_j^* + \epsilon_j| > \tau_k\right\}(b_j^* + \epsilon_j + \mathbf{x}_j^\top\boldsymbol{\lambda}) - b_j^*) \\
&= \sum_j (\boldsymbol{\lambda}^\top\mathbf{x}_j)^2\mathbb{I}\left\{|\mathbf{x}_j^\top\boldsymbol{\lambda} + b_j^* + \epsilon_j| > \tau_k\right\} + \sum_j (\boldsymbol{\lambda}^\top\mathbf{x}_j)(\mathbb{I}\left\{|\mathbf{x}_j^\top\boldsymbol{\lambda} + b_j^* + \epsilon_j| > \tau_k\right\}(b_j^* + \epsilon_j) - b_j^*) \\
&\leq 20(k\log n/k + \sqrt{dn\log n})\|\boldsymbol{\lambda}\|^2 + \sum_j (\boldsymbol{\lambda}^\top\mathbf{x}_j)(\mathbb{I}\left\{|\mathbf{x}_j^\top\boldsymbol{\lambda} + b_j^* + \epsilon_j| > \tau_k\right\}(b_j^* + \epsilon_j) - b_j^*), \tag{8}
\end{aligned}
$$

where the last inequality follows from Theorem 10 and hold with probability $\geq 1 - 1/n^d$.

Now to bound the second term above, our approach is to show that each of the $j$-th term is small in expectation and then use tail bounds to obtain the final bound. Unfortunately, as $\boldsymbol{\lambda}$ and $\tau_k$ can depend on random variables $X$ and $\epsilon$, we cannot bound expectation as well as apply Hoeffding style tail bounds directly.

Instead, we take the standard approach of using $\epsilon$-nets and union bounds. That is, we form $\gamma_{\boldsymbol{\lambda}}$-net over the set $B := \mathcal{B}_2(\mathbf{0}, \frac{\sigma}{100})$ (i.e. the ball of radius $\frac{\sigma}{100} \geq \|\boldsymbol{\lambda}\|$ centered at the origin in $d$-dimensions) and similarly a unit dimensional $\gamma_\tau$-net over $[\sigma/2\sqrt{\log(n/k)}, \; 30\sigma\sqrt{\log(n/k)}]$; we will provide the justification for selecting the above given range for $\tau$ below.

Recall that we hard-threshold $k = 2k^*$ entries of $X^\top\boldsymbol{\lambda}+\boldsymbol{\epsilon}+\mathbf{b}^*$. Moreover, $|supp(\mathbf{b}^*)| \leq k^*$. Now, let $S = \{j \text{ s.t. } |x_j^\top\boldsymbol{\lambda}+\epsilon_j| \geq \tau_k$. Note that $|S| \geq k-k^*$ as $\tau_k$ ensures that top-$k$ elements of $X^\top\boldsymbol{\lambda}+\boldsymbol{\epsilon}+\mathbf{b}^*$ are selected. Similarly, define $\widehat{S} = \{j \text{ s.t. } |x_j^\top\boldsymbol{\lambda}+\epsilon_j| < \tau_k$. Again, note that $|\widehat{S}| \leq n - k + k^*$ as $\tau_k$ ensures that *only* $k$ elements of $X^\top\boldsymbol{\lambda} + \boldsymbol{\epsilon} + \mathbf{b}^*$ are selected. Hence,

$$\frac{1}{|\widehat{S}|}\sum_{j\in\widehat{S}}(x_j^\top\boldsymbol{\lambda} + \epsilon_j)^2 \leq \tau_k^2 \leq \frac{1}{|S|}\sum_{j\in S}(x_j^\top\boldsymbol{\lambda} + \epsilon_j)^2. \tag{9}$$

Using the fact that $k \leq n/10000$ and using SSC/SSS bounds with $d < n/10000$ (apply Theorem 10 part 1 with $k - k^*$ and part 2 with $n - k + k^*$), we get with probability at least $1 - \frac{1}{n^6}$, $\tau_k \in [.5\sigma\sqrt{\log n/k}, 30\sigma\sqrt{\log n/k}]$. Note that this result holds uniformly for any $\boldsymbol{\lambda} \in \mathcal{B}_2(\mathbf{0}, \frac{\sigma}{100})$ as it simply uses the SSC/SSS properties of the matrix $X$.

Now, let $\widetilde{\boldsymbol{\lambda}}$ and $\widetilde{\tau}$ be the closest point from the $\gamma_{\boldsymbol{\lambda}}$-net to $\boldsymbol{\lambda}$ and the $\gamma_\tau$-net to $\tau_k$, respectively. Now,

$$\left|\sum_j \mathbb{I}\left\{|\mathbf{x}_j^\top\boldsymbol{\lambda} + \epsilon_j + b_j^*| \geq \tau_k\right\} - \mathbb{I}\left\{|\mathbf{x}_j^\top\widetilde{\boldsymbol{\lambda}} + \epsilon_j + b_j^*| \geq \widetilde{\tau}\right\}\right| \leq \sum_j \mathbb{I}\left\{|\mathbf{x}_j^\top\widetilde{\boldsymbol{\lambda}} + \epsilon_j + b_j^* - \widetilde{\tau}| \leq (\gamma_\tau + \|\mathbf{x}_j\|\gamma_{\boldsymbol{\lambda}})\right\}. \tag{10}$$

Now let us denote $\nu_j = \mathbf{x}_j^\top\widetilde{\boldsymbol{\lambda}} + \epsilon_j + b_j^* \sim \mathcal{N}(b_j^*, \|\boldsymbol{\lambda}\|^2 + \sigma^2)$. Note that $\nu_j$ are independent of each other (recall that $\mathbf{b}^*$ is generated independently of $\mathbf{x}_i$ and $\boldsymbol{\lambda}$ is a fixed vector). Also, let $\gamma = \gamma_\tau + \|\mathbf{x}_j\|\gamma_{\boldsymbol{\lambda}}$. Recall that since the Gaussian distribution has a density function that always takes values less than unity at every point, for $g \sim \mathcal{N}(\mu, 1)$, $\Pr(|g - \tau| \leq \zeta) \leq 2\zeta$ for all $\tau, \mu$ and for all $\zeta > 0$. Hence, for a fixed $\widetilde{\boldsymbol{\lambda}}$ and $\widetilde{\tau}$, we have:

$$\Pr\left(\sum_j \mathbb{I}\{|\nu_j - \widetilde{\tau}| \leq \gamma\} \geq 4d\right) = \Pr\left(\sum_j \mathbb{I}\left\{\left|\frac{\nu_j}{\widetilde{\sigma}} - \frac{\widetilde{\tau}}{\widetilde{\sigma}}\right| \leq \frac{\gamma}{\widetilde{\sigma}}\right\} \geq 4d\right) \leq \binom{n}{4d}\left(\frac{2\gamma}{\widetilde{\sigma}}\right)^{4d} \leq \binom{n}{4d}\left(\frac{2\gamma}{\sigma}\right)^{4d}. \tag{11}$$

In the above $\widetilde{\sigma} = \sqrt{\sigma^2 + \|\boldsymbol{\lambda}\|_2^2} \geq \sigma$. At this point, we recall that using standard results on covering numbers [9] and by setting $\gamma_\tau = \mathcal{O}\left(\frac{\sigma}{n^3}\right)$, $\gamma_{\boldsymbol{\lambda}} = \frac{\sigma}{2000 dn^3 \log n}$, we know that the net on $\lambda$ values needs at most $(\frac{4\sigma}{100\gamma_{\boldsymbol{\lambda}}})^d$ elements and the net on $\tau$ values needs at most $\frac{2.5\sigma\sqrt{\log(n/k)}}{\gamma_\tau}$ elements. Thus, using a union bound, we have *for all* $\widetilde{\boldsymbol{\lambda}}$ in the $\gamma_{\boldsymbol{\lambda}}$-net of $B$ and $\widetilde{\tau} \in \gamma_\tau$-net of $[.5\sigma\sqrt{\log(n/k)}, 30\sigma\sqrt{\log(n/k)}]$:

$$\Pr\left(\sum_j \mathbb{I}\{|\nu_j - \widetilde{\tau}| \leq \gamma\} \geq 4d\right) \leq \binom{n}{4d}\left(\frac{2\gamma}{\sigma}\right)^{4d} \cdot \left(\frac{4\sigma}{100\gamma_{\boldsymbol{\lambda}}}\right)^d \cdot \left(\frac{2.5\sigma\sqrt{\log(n/k)}}{\gamma_\tau}\right) \leq \frac{2\sqrt{\log n}}{n^{5d}}, \tag{12}$$

where the last inequality follows by setting $\gamma_\tau = \mathcal{O}\left(\frac{\sigma}{n^3}\right)$, $\gamma_{\boldsymbol{\lambda}} = \frac{\sigma}{2000 dn^3 \log n}$. These bounds were set as such since $\|\boldsymbol{\lambda}\|_2 \leq \sigma/100$, and by using standard tail bounds on Chi-squared random variables, with probability at least $1 - 1/n^d$, we have $\max_i \|\mathbf{x}_i\|_2 \leq 20d \log n$. Using (10) with the above bound, we get that $w.p. \geq 1 - \frac{1}{n^d}$:

$$\left|\sum_j \mathbb{I}\left\{|\mathbf{x}_j^\top\boldsymbol{\lambda} + \epsilon_j + b_j^*| \geq \tau_k\right\} - \mathbb{I}\left\{|\mathbf{x}_j^\top\widetilde{\boldsymbol{\lambda}} + \epsilon_j + b_j^*| \geq \widetilde{\tau}\right\}\right| \leq 4d. \tag{13}$$

Also let $R = \left\{j, \text{ s.t., } \mathbb{I}\left\{|\mathbf{x}_j^\top\boldsymbol{\lambda} + \epsilon_j + b_j^*| \geq \tau_k\right\} \neq \mathbb{I}\left\{|\mathbf{x}_j^\top\widetilde{\boldsymbol{\lambda}} + \epsilon_j + b_j^*| \geq \widetilde{\tau}\right\}\right\}$. Above bound shows that $|R| \leq 4d$ with high probability.

Now,

$$\sum_{j\in R}|\boldsymbol{\lambda}^\top\mathbf{x}_j||b_j^* + \epsilon_j| \leq \sum_{j\in R}|\boldsymbol{\lambda}^\top\mathbf{x}_j|(|\boldsymbol{\lambda}^\top\mathbf{x}_j| + \widetilde{\tau} + \gamma) \overset{\zeta_1}{\leq} \sum_{j\in R}(\boldsymbol{\lambda}^\top\mathbf{x}_j)^2 + \sqrt{|R|}(\widetilde{\tau} + \gamma)\sqrt{\sum_{j\in R}(\boldsymbol{\lambda}^\top\mathbf{x}_j)^2}$$

$$\overset{\zeta_2}{\leq} 20(d \log n/d + \sqrt{dn \log n})\|\boldsymbol{\lambda}\|^2 + 160\sigma(d + \sqrt{dn}) \log n \cdot \|\boldsymbol{\lambda}\|, \tag{14}$$

where $\zeta_1$ follows from the fact that for all $j \in R$, $|b_j^* + \epsilon_j + \boldsymbol{\lambda}^\top \mathbf{x}_j - \widetilde{\tau}| \le \gamma$ and by using Cauchy-Schwarz inequality. $\zeta_2$ follows from SSS condition (Theorem 10), as well as bound on $\tau$ and $\gamma$ (given above).

Using (8) and (14), we get (w.p. $\ge 1 - 1/n^d$), for any $\boldsymbol{\lambda} \in \mathcal{B}_2(\mathbf{0}, \frac{\sigma}{100})$:

$$\boldsymbol{\lambda}^\top X \mathbf{z} \le 20((k+d)\log n/d + 2\sqrt{dn \log n}) \|\boldsymbol{\lambda}\|^2 + 160\sigma(d + \sqrt{dn}) \cdot \log n \cdot \|\boldsymbol{\lambda}\|$$
$$+ \sum_j (\boldsymbol{\lambda}^\top \mathbf{x}_j) \left( \mathbb{I}\left\{ |\mathbf{x}_j^\top \widetilde{\boldsymbol{\lambda}} + b_j^* + \epsilon_j| > \widetilde{\tau} \right\} (b_j^* + \epsilon_j) - b_j^* \right). \quad (15)$$

We now analyze the last term in the above expression:

$$\sum_j (\boldsymbol{\lambda}^\top \mathbf{x}_j) \left( \mathbb{I}\left\{ |\mathbf{x}_j^\top \widetilde{\boldsymbol{\lambda}} + b_j^* + \epsilon_j| > \widetilde{\tau} \right\} (b_j^* + \epsilon_j) - b_j^* \right) \overset{\zeta_1}{\le} \sum_j (\widetilde{\boldsymbol{\lambda}}^\top \mathbf{x}_j) \left( \mathbb{I}\left\{ |\mathbf{x}_j^\top \widetilde{\boldsymbol{\lambda}} + b_j^* + \epsilon_j| > \widetilde{\tau} \right\} (b_j^* + \epsilon_j) - b_j^* \right)$$

$$+ \gamma_{\boldsymbol{\lambda}} \sqrt{d} \sqrt{\log n} \left( \sum_j \left| \left( \mathbb{I}\left\{ |\mathbf{x}_j^\top \widetilde{\boldsymbol{\lambda}} + b_j^* + \epsilon_j| > \widetilde{\tau} \right\} - 1 \right) b_j^* \right| + |\epsilon_j| \right),$$

$$\overset{\zeta_2}{\le} \sum_j (\widetilde{\boldsymbol{\lambda}}^\top \mathbf{x}_j) \left( \mathbb{I}\left\{ |\mathbf{x}_j^\top \widetilde{\boldsymbol{\lambda}} + b_j^* + \epsilon_j| > \widetilde{\tau} \right\} (b_j^* + \epsilon_j) - b_j^* \right) + \gamma_{\boldsymbol{\lambda}} \sqrt{d} \sqrt{\log n} \cdot n \cdot \sqrt{d} \sigma \sqrt{\log n}$$

$$\overset{\zeta_3}{\le} \sum_j (\widetilde{\boldsymbol{\lambda}}^\top \mathbf{x}_j) \left( \mathbb{I}\left\{ |\mathbf{x}_j^\top \widetilde{\boldsymbol{\lambda}} + b_j^* + \epsilon_j| > \widetilde{\tau} \right\} (b_j^* + \epsilon_j) - b_j^* \right) + \frac{\sigma}{2000 n^2}, \quad (16)$$

where $(\zeta_1)$ follows by using the fact that $\left\| \boldsymbol{\lambda} - \tilde{\boldsymbol{\lambda}} \right\|_2 \le \gamma_{\boldsymbol{\lambda}}$ by definition, $(\zeta_2)$ follows from bounds on $\epsilon_j$, and $(\zeta_3)$ follows from the setting of $\gamma_{\boldsymbol{\lambda}} = \frac{\sigma}{2000 d n^3 \log n}$.

Now, using a union bound over all the $(80 d n^3 \log n)^d$ elements of the net over $\boldsymbol{\lambda}$ values and all $2.5 n^3 \sqrt{\log n}$ elements of the net over $\tau$ values on top of the result in Lemma 13, we have, w.p. $1 - 1/n^d$, *for all* $\widetilde{\boldsymbol{\lambda}} \in \gamma_{\boldsymbol{\lambda}}$-net of $B$ and for all $\widetilde{\tau} \in \gamma_\tau$-net of $[.5\sigma\sqrt{\log n/k}, 30\sigma\sqrt{\log n/k}]$:

$$\sum_j (\widetilde{\boldsymbol{\lambda}}^\top \mathbf{x}_j) \left( \mathbb{I}\left\{ |\mathbf{x}_j^\top \widetilde{\boldsymbol{\lambda}} + b_j^* + \epsilon_j| > \widetilde{\tau} \right\} (b_j^* + \epsilon_j) - b_j^* \right) \le \left( 0.4n\sqrt{\log \alpha} \exp(-\frac{\log \alpha}{33}) + 1.62 \frac{n}{\alpha} \log(\alpha) \right) \|\boldsymbol{\lambda}\|_2^2$$

$$+ 8\sigma \|\boldsymbol{\lambda}\| \log n \cdot \sqrt{100 n d \log n}, \quad (17)$$

where $\alpha = n/k$.

Finally, using (15), (16), and (17), and the fact that $\alpha = n/(2k^*) \ge 10000$ and $d \le n/10000$, we get with probability at least $\ge 1 - 1/n^{10}$:

$$\boldsymbol{\lambda}^\top X \mathbf{z} \le .8n\|\boldsymbol{\lambda}\|^2 + 200\sigma\sqrt{n \cdot d} \cdot \log^2 n \cdot \|\boldsymbol{\lambda}\|. \quad (18)$$

We now consider the second term from (7). Let $\mathbf{v}$ be any unit vector such that $\mathbf{v}^\top \boldsymbol{\lambda} = 0$. Note that,

$$\mathbf{v}^T X \mathbf{z} = \widetilde{\mathbf{v}}^\top X \mathbf{z} - (\widetilde{\mathbf{v}} - \mathbf{v})^\top X \mathbf{z}, \quad (19)$$

where $\widetilde{\mathbf{v}}$ is the closest point to $\mathbf{v}$ in an $\nu$-net over $S^{d-1}$ such that each point over the net is orthogonal to $\widetilde{\boldsymbol{\lambda}}$; recall that $\widetilde{\boldsymbol{\lambda}}$ is the closest point to $\boldsymbol{\lambda}$ over $\gamma_{\boldsymbol{\lambda}}$ net over $B$.

Now,

$$\widetilde{\mathbf{v}}^\top X \mathbf{z} = \sum_j (\widetilde{\mathbf{v}}^\top \mathbf{x}_j)(\mathbb{I}\left\{ |\boldsymbol{\lambda}^\top \mathbf{x}_j + b_j^* + \epsilon_j| \ge \tau_k \right\} (b_j^* + \epsilon_j) - b_j^*), \quad (20)$$

$$= \sum_j (\widetilde{\mathbf{v}}^\top \mathbf{x}_j)(\mathbb{I}\left\{ |\widetilde{\boldsymbol{\lambda}}^\top \mathbf{x}_j + b_j^* + \epsilon_j| \ge \widetilde{\tau} \right\} (b_j^* + \epsilon_j) - b_j^*)$$

$$+ \sum_j (\widetilde{\mathbf{v}}^\top \mathbf{x}_j)(\mathbb{I}\left\{ |\boldsymbol{\lambda}^\top \mathbf{x}_j + b_j^* + \epsilon_j| \ge \tau_k \right\} - \mathbb{I}\left\{ |\widetilde{\boldsymbol{\lambda}}^\top \mathbf{x}_j + b_j^* + \epsilon_j| \ge \widetilde{\tau} \right\})(b_j^* + \epsilon_j). \quad (21)$$

Now, using the fact that,

$$\left| \mathbb{I}\left\{|\boldsymbol{\lambda}^\top \mathbf{x}_j + b_j^* + \epsilon_j| \geq \tau_k\right\} - \mathbb{I}\left\{|\widetilde{\boldsymbol{\lambda}}^\top \mathbf{x}_j + b_j^* + \epsilon_j| \geq \widetilde{\tau}\right\}\right| \leq \mathbb{I}\left\{\widetilde{\boldsymbol{\lambda}}^\top \mathbf{x}_j + b_j^* + \epsilon_j - \widetilde{\tau}| \leq \gamma_\tau + \|\mathbf{x}_j\|\gamma_{\boldsymbol{\lambda}}\right\},$$

as well as using (13), we get (w.p. $\geq 1 - 1/n^d$):

$$\sum_j (\widetilde{\mathbf{v}}^\top \mathbf{x}_j)(\mathbb{I}\left\{|\boldsymbol{\lambda}^\top \mathbf{x}_j + b_j^* + \epsilon_j| \geq \tau_k\right\} - \mathbb{I}\left\{|\widetilde{\boldsymbol{\lambda}}^\top \mathbf{x}_j + b_j^* + \epsilon_j| \geq \widetilde{\tau}\right\})(b_j^* + \epsilon_j) \leq \sum_{j \in R} |\widetilde{\mathbf{v}}^\top \mathbf{x}_j|(|\widetilde{\boldsymbol{\lambda}}^\top \mathbf{x}_j| + \widetilde{\tau} + \gamma_\tau + \|\mathbf{x}_j\|\gamma_{\boldsymbol{\lambda}}),$$

$$\overset{\zeta_1}{\leq} 2\|\boldsymbol{\lambda}\| \sum_{j \in R} (\widetilde{\mathbf{v}}^\top \mathbf{x}_j)^2 + 2\frac{1}{\|\boldsymbol{\lambda}\|} \sum_{j \in R} (\widetilde{\boldsymbol{\lambda}}^\top \mathbf{x}_j)^2 + 30\sigma \log n \sum_{j \in R} |\widetilde{\mathbf{v}}^\top \mathbf{x}_j|,$$

$$\overset{\zeta_2}{\leq} (d + \sqrt{dn})\sigma + 100\sigma d \log n, \tag{22}$$

where $R = \left\{j, \text{ s.t., } \sum_j \mathbb{I}\left\{|\mathbf{x}_j^\top \boldsymbol{\lambda} + \epsilon_j + b_j^*| \geq \tau_k\right\} \neq \mathbb{I}\left\{|\mathbf{x}_j^\top \widetilde{\boldsymbol{\lambda}} + \epsilon_j + b_j^*| \geq \widetilde{\tau}\right\}\right\}$. $\zeta_1$ follows from $\widetilde{\tau} \leq 30\sigma\sqrt{\log n}$. $\zeta_2$ follows from Theorem 10 along with bound on $|R| \leq 4d$ obtained using (13).

Now consider the first term of (21). Using an argument similar to the one used in Lemma 13, we get (w.p. $\geq 1 - \exp(-n/10)$):

$$|(\widetilde{\mathbf{v}}^\top \mathbf{x}_j)(\mathbb{I}\left\{|\widetilde{\boldsymbol{\lambda}}^\top \mathbf{x}_j + b_j^* + \epsilon_j| \geq \widetilde{\tau}\right\}(b_j^* + \epsilon_j) - b_j^*)| \leq 8\sigma \log n. \tag{23}$$

Now since $\mathbf{v} \perp \boldsymbol{\lambda}$ and $\mathbf{x}_j$ is a standard multivariate Gaussian random variable, $\boldsymbol{\lambda}^\top \mathbf{x}_j$ and $\mathbf{v}^\top \mathbf{x}_j$ are independent random variables. Moreover $\mathbb{E}\left[\mathbf{v}^\top \mathbf{x}_j\right] = 0$. Thus we have

$$\mathbb{E}\left[\sum_j (\mathbf{v}^\top \mathbf{x}_j)(\mathbb{I}\left\{|\boldsymbol{\lambda}^\top \mathbf{x}_j + b_j^* + \epsilon_j| \geq \widetilde{\tau}\right\}(b_j^* + \epsilon_j) - b_j^*)\right] = 0$$

Using simple manipulations, we can then show

$$\mathbb{E}\left[\sum_j (\widetilde{\mathbf{v}}^\top \mathbf{x}_j)(\mathbb{I}\left\{|\widetilde{\boldsymbol{\lambda}}^\top \mathbf{x}_j + b_j^* + \epsilon_j| \geq \widetilde{\tau}\right\}(b_j^* + \epsilon_j) - b_j^*)\right] \leq n(\gamma_\tau + \gamma_{\boldsymbol{\lambda}} \cdot d \log n),$$

where the right hand side is vanishingly small since we set $\gamma_\tau$ and $\gamma_{\boldsymbol{\lambda}}$ to values of the order of $n^{-3}$. Even the above can be avoided and the right hand side made absolutely zero by simply expanding the $\nu$-net over $\mathbf{v}$ vectors so that for every pair of $(\mathbf{v}, \boldsymbol{\lambda})$ that is orthogonal, we are able to find a pair $(\tilde{\mathbf{v}}, \tilde{\boldsymbol{\lambda}})$ that are not only in close proximity to the original pair but we also have $\tilde{\mathbf{v}} \perp \tilde{\boldsymbol{\lambda}}$. This would require a quadratic increase in the number of points in the net which can only increase the constants in the expressions by a small quantity.

Thereafter, using the Hoeffding's inequality with union bound as well (19), (21), and (23), we get w.p. $\geq 1 - 1/n^{d-1}$, *for all* $\boldsymbol{\lambda}$ and *for all* $\mathbf{v} \in S^{d-1}$ s.t. $\mathbf{v} \perp \boldsymbol{\lambda}$:

$$|\mathbf{v}^T X \mathbf{z}| \leq 100\sigma(d + \sqrt{d \cdot n}) \log^2 n. \tag{24}$$

The result now follows by combining (7), (18), and (24) and using $\sqrt{a+b} \leq \sqrt{a} + \sqrt{b}$. $\qquad \square$

**Lemma 13.** *Let* $\mathbf{x}_j \sim \mathcal{N}(0, I) \in \mathbb{R}^d$ *for all* $1 \leq j \leq n$ *and* $\epsilon_j \sim \mathcal{N}(0, \sigma^2)$. *Let* $b^*$ *be* $k^*$ *sparse and* $k = 2k^*$. *Let* $d < n/10000$, $k < n/10000$ *and* $\alpha = n/k$. *Let* $\boldsymbol{\lambda} \in \mathbb{R}^d$ *be a* fixed *vector with* $\|\boldsymbol{\lambda}\| \leq \sigma/100$ *and* $\tau \in [.5\sigma \log(n/k), 2\sigma \log(n/k)]$ *be a fixed constant. Then, the following holds (w.p.* $\geq 1 - \delta - \exp(-n/10)$):

$$\sum_j (\boldsymbol{\lambda}^\top \mathbf{x}_j)\left(\mathbb{I}\left\{|\mathbf{x}_j^\top \boldsymbol{\lambda} + b_j^* + \epsilon_j| > \tau\right\}(b_j^* + \epsilon_j) - b_j^*\right) \leq \left(0.4n\sqrt{\log \alpha}\exp(-\frac{\log \alpha}{33}) + 1.62\frac{n}{\alpha}\log(\alpha)\right)\|\boldsymbol{\lambda}\|_2^2$$

$$+ 8\sigma\|\boldsymbol{\lambda}\|\log n\sqrt{n\log\frac{n}{\delta}}.$$

*Proof.* Let us define $\mathbf{x}_j^\top \boldsymbol{\lambda} := h_j, a_j = b_j^* + \epsilon_j$. We note that $h_j \sim N(0, \|\boldsymbol{\lambda}\|_2^2)$ and $a_j \sim N(b_j^*, \sigma^2)$. Then, we are interested in:

$$\sum_j (\boldsymbol{\lambda}^\top \mathbf{x}_j)\left(\mathbb{I}\left\{|\mathbf{x}_j^\top \boldsymbol{\lambda} + b_j^* + \epsilon_j| > \tau\right\}(b_j^* + \epsilon_j) - b_j^*\right) = \sum_j r_j, \tag{25}$$

where
$$r_j = \mathbb{I}\{|h_j + a_j| > \tau\} h_j a_j - h_j b_j^*.$$

Note that $\mathbb{E}\left[\sum_j h_j b_j^*\right] = 0$ since $b_j^*$ is independent of $h_j$ and $\boldsymbol{\lambda}$ is fixed. Hence,

$$\mathbb{E}\left[\sum_j r_j\right] = \mathbb{E}\left[\sum_j \mathbb{I}\{|h_j + a_j| > \tau\} h_j a_j\right]. \tag{26}$$

Now, using distribution of $h_j$ and $a_j$, we have:

$$\mathbb{E}\left[\mathbb{I}\{|h_j + a_j| > \tau\} h_j a_j | a_j\right] = \frac{1}{\sqrt{2\pi}\|\boldsymbol{\lambda}\|_2} \int_{-\infty}^{-\tau-a_j} a_j h_j \exp\left(-\frac{h_j^2}{2\|\boldsymbol{\lambda}\|_2^2}\right) dh_j + \frac{1}{\sqrt{2\pi}\|\boldsymbol{\lambda}\|_2} \int_{\tau-a_j}^{\infty} a_j h_j \exp\left(-\frac{h_j^2}{2\|\boldsymbol{\lambda}\|_2^2}\right) dh_j$$

$$= \frac{\|\boldsymbol{\lambda}\|_2}{\sqrt{2\pi}} a_j \left(\exp\left(-\frac{(\tau-a_j)^2}{2\|\boldsymbol{\lambda}\|_2^2}\right) - \exp\left(-\frac{(\tau+a_j)^2}{2\|\boldsymbol{\lambda}\|_2^2}\right)\right).$$

Since $h_j$ and $a_j$ are independent $\mathbb{E}\left[\mathbb{I}\{|h_j + a_j| > \tau\} h_j a_j\right] = \mathbb{E}_{a_j}\left[\mathbb{E}_{h_j}\left[\mathbb{I}\{|h_j + a_j| > \tau\} h_j a_j | a_j\right]\right]$. Therefore,

$$\mathbb{E}\left[\mathbb{I}\{|h_j + a_j| > \tau\} h_j a_j\right] = \frac{\|\boldsymbol{\lambda}\|_2^2}{\sqrt{2\pi}\|\boldsymbol{\lambda}\|_2} \frac{1}{\sqrt{2\pi}\sigma} \int_{-\infty}^{\infty} a_j \exp\left(-\frac{(\tau-a_j)^2}{2\|\boldsymbol{\lambda}\|_2^2}\right) \exp\left(-\frac{(a_j - b_j^*)^2}{2\sigma^2}\right) da_j$$

$$- \frac{\|\boldsymbol{\lambda}\|_2^2}{\sqrt{2\pi}\|\boldsymbol{\lambda}\|_2} \frac{1}{\sqrt{2\pi}\sigma} \int_{-\infty}^{\infty} a_j \exp\left(-\frac{(\tau+a_j)^2}{2\|\boldsymbol{\lambda}\|_2^2}\right) \exp\left(-\frac{(a_j - b_j^*)^2}{2\sigma^2}\right) da_j$$

$$= \frac{1}{\sqrt{2\pi}} \frac{\|\boldsymbol{\lambda}\|_2^2}{\sqrt{\|\boldsymbol{\lambda}\|_2^2 + \sigma^2}} \exp\left(-\frac{1}{2}\left(\frac{\tau^2}{\|\boldsymbol{\lambda}\|_2^2} + \frac{b_j^{*2}}{\sigma^2}\right)\right) \frac{\frac{\tau}{\|\boldsymbol{\lambda}\|_2^2} + \frac{b_j^*}{\sigma^2}}{\frac{1}{\|\boldsymbol{\lambda}\|_2^2} + \frac{1}{\sigma^2}} \exp\left(\frac{1}{2}\frac{\left(\frac{\tau}{\|\boldsymbol{\lambda}\|_2^2} + \frac{b_j^*}{\sigma^2}\right)^2}{\frac{1}{\|\boldsymbol{\lambda}\|_2^2} + \frac{1}{\sigma^2}}\right)$$

$$+ \frac{1}{\sqrt{2\pi}} \frac{\|\boldsymbol{\lambda}\|_2^2}{\sqrt{\|\boldsymbol{\lambda}\|_2^2 + \sigma^2}} \exp\left(-\frac{1}{2}\left(\frac{\tau^2}{\|\boldsymbol{\lambda}\|_2^2} + \frac{b_j^{*2}}{\sigma^2}\right)\right) \frac{\frac{\tau}{\|\boldsymbol{\lambda}\|_2^2} - \frac{b_j^*}{\sigma^2}}{\frac{1}{\|\boldsymbol{\lambda}\|_2^2} + \frac{1}{\sigma^2}} \exp\left(\frac{1}{2}\frac{\left(\frac{-\tau}{\|\boldsymbol{\lambda}\|_2^2} + \frac{b_j^*}{\sigma^2}\right)^2}{\frac{1}{\|\boldsymbol{\lambda}\|_2^2} + \frac{1}{\sigma^2}}\right)$$

$$\leq 2\frac{\|\boldsymbol{\lambda}\|_2^2}{\sqrt{2\pi}} \frac{\tau + \frac{|b_j^*|\|\boldsymbol{\lambda}\|_2^2}{\sigma^2}}{\sigma} \exp\left(\frac{-1}{2}\frac{(|b_j^*| - \tau)^2}{\sigma^2 + \|\boldsymbol{\lambda}\|_2^2}\right), \tag{27}$$

where the above inequalities follow from straightforward calculations.

Note that $\|\boldsymbol{\lambda}\|_2^2 \leq \frac{\sigma^2}{10^4}$. Let $\alpha := \frac{n}{k}$. Now, consider the following three cases for a fixed $\tau \in (\frac{1}{2}\sqrt{\log(\alpha)}, 2\sqrt{\log(\alpha)})$:

- $|b_j^*| \leq \tau/2$

$$\mathbb{E}[r_j] \leq \|\boldsymbol{\lambda}\|_2^2 \frac{2}{\sqrt{2\pi}} \frac{\tau + \frac{\tau}{2\cdot10^4}}{\sigma} \exp\left(\frac{-1}{2.02}\frac{(|b_j^*| - \tau)^2}{\sigma^2}\right) \overset{\zeta_1}{\leq} \|\boldsymbol{\lambda}\|_2^2 0.8\frac{\tau}{\sigma} \exp\left(\frac{-1}{8.08}\frac{\tau^2}{\sigma^2}\right)$$

  where $\zeta_1$ follows from using the fact that $|b_j^*| \leq \tau/2$.

- $\tau/2 \leq |b_j^*| \leq 2\tau$

$$\mathbb{E}[r_j] \leq \|\boldsymbol{\lambda}\|_2^2 \frac{2}{\sqrt{2\pi}} \frac{\tau + \frac{2\tau}{10^4}}{\sigma} \exp\left(\frac{-1}{2.02}\frac{(|b_j^*| - \tau)^2}{\sigma^2}\right) \overset{\zeta_1}{\leq} \|\boldsymbol{\lambda}\|_2^2 0.8\frac{\tau}{\sigma}$$

  where $\zeta_1$ follows from using $\exp\left(\frac{-1}{2.02}\frac{(|b_j^*| - \tau)^2}{\sigma^2}\right) \leq 1$.

- $|b_j^*| \geq 2\tau$

$$\mathbb{E}[r_j] \leq \|\boldsymbol{\lambda}\|_2^2 \frac{2}{\sqrt{2\pi}} \frac{0.5001|b_j^*|}{\sigma} \exp\left(\frac{-1}{2.02}\frac{(|b_j^*| - \tau)^2}{\sigma^2}\right) \overset{\zeta_1}{\leq} \|\boldsymbol{\lambda}\|_2^2 0.4\frac{|b_j^*|}{\sigma} \exp\left(\frac{-1}{8.08}\frac{b_j^{*2}}{\sigma^2}\right)$$

  where $\zeta_1$ follows from using $|b_j^*| \geq 2\tau$.

Recall that $\mathbf{b}^*$ has at most $k^*$ non-zeros, hence at most $k^*$ elements can belong to case 2 and 3 described above. Combining the above given cases, we have:

$$\mathbb{E}\left[\sum_j r_j\right] \leq \left((n-k^*)\max_{\beta \geq \frac{2\tau}{\sigma}}\left[0.4\beta\exp\left(\frac{-\beta^2}{8.08}\right), 0.8\frac{\tau}{\sigma}\exp\left(\frac{-1}{8.08}\frac{\tau^2}{\sigma^2}\right)\right] + 0.8\frac{\tau}{\sigma}\cdot k^*\right)\|\boldsymbol{\lambda}\|_2^2$$

$$\overset{\zeta_1}{\leq}\left((n-k^*)\max_{\beta \geq \frac{2\tau}{\sigma}}\left[0.4\beta\exp\left(\frac{-\beta^2}{8.08}\right), 0.8\frac{\tau}{\sigma}\exp\left(\frac{-1}{8.08}\frac{\tau^2}{\sigma^2}\right)\right] + 1.62\frac{n}{\alpha}\log(\alpha)\right)\|\boldsymbol{\lambda}\|_2^2$$

$$\overset{\zeta_2}{\leq}\left(0.4n\sqrt{\log\alpha}\exp(-\frac{\log\alpha}{33}) + 1.62\frac{n}{\alpha}\log(\alpha)\right)\|\boldsymbol{\lambda}\|_2^2, \tag{28}$$

where $\zeta_1$ follows from the fact that $\tau \leq 30\sigma\sqrt{\log(\alpha)}$. $\zeta_2$ follows from maximizing $x\exp\left(-\frac{x^2}{8.08}\right)$ in the interval $(0.5\sqrt{\log(\alpha)}, 2\sqrt{\log(\alpha)})$ for $\tau$ and $(\sqrt{\log(\alpha)}, \infty)$ for $\beta$.

Above given equation bounds the expected values of $\sum_j r_j$. We now bound the deviation of $\sum_j r_j$ from its expected value. For this, we first consider $r_j = \mathbb{I}\{|h_j + a_j| > \tau\}h_j a_j - h_j b_j^*$. Here, we consider two cases:

1. $|h_j + a_j| > \tau$: in this case, $r_j = h_j \epsilon_j$. Moreover, w.p. $\geq 1 - \exp(-n/10)$, $|h_j| \leq 2\|\boldsymbol{\lambda}\|\sqrt{\log n}$, and $|\epsilon_j| \leq 2\sigma\sqrt{\log n}$. Hence, $|r_j| \leq 4\|\boldsymbol{\lambda}\|\sigma\log n$.
2. $|h_j + a_j| \leq \tau$: in this case, $|b_j^*| \leq \tau + |h_j| + |\epsilon_j| \leq \tau + 2(\|\boldsymbol{\lambda}\| + \sigma)\sqrt{\log n}$. Moreover, $r_j = -h_j b_j^*$, i.e., $|r_j| \leq 8\sigma\|\boldsymbol{\lambda}\|\log n$.

Hence, using Hoeffding's bound, we get (w.p. $\geq 1 - \delta - \exp(-n/10)$):

$$\left|\sum_j r_j - \mathbb{E}\left[r_j\right]\right| \leq 8\sigma\|\boldsymbol{\lambda}\|\log n\sqrt{n\log\left(\frac{2}{\delta}\right)}.$$

The result now follows by combining the above observation with (28). $\qquad\square$

## Footnotes

[3] $\frac{1}{\sqrt{2\pi}}\int_x^\infty e^{-t^2/2}dt \leq \frac{1}{\sqrt{2\pi}}\int_x^\infty \frac{t}{x}e^{-t^2/2}dt = \frac{1}{x\sqrt{2\pi}}e^{-x^2/2}$