[Reviews · NeurIPS 2017]

Reviewer 1



++++++++ Summary: ++++++++ The paper presents a provably consistent, polynomial-time algorithm (that they call consistent robust regression, or CRR) for linear regression with corrupted samples under the oblivious adversary model. The algorithm is a simple iterative hard thresholding (IHT) procedure, and the authors show that this algorithm exhibits linear convergence. The analysis is supported by some representative synthetic numerical results. ++++++++ Strengths: ++++++++ Quality: Somewhat surprisingly, this method appears to be the first consistent procedure for robust linear regression, i.e., the parameter estimation error vanishes as the number of samples tends to infinity. Other estimation procedures (such as LASSO-type methods) seem to only provide solutions with error comparable to the noise level, even in the large sample limit. Clarity: The paper is nicely written. The authors succinctly state their contributions, put them in the context of existing literature, and provide high level intuition as to what’s happening behind the scenes. ++++++++++ Weaknesses: ++++++++++ Novelty/Significance: The reformulation of the robust regression problem (Eq 6 in the paper) shows that robust regression is reducible to standard k-sparse recovery. Therefore, the proposed CRR algorithm is basically the well-known IHT algorithm (with a modified design matrix), and IHT has been (re)introduced far too many times in the literature to count. The proofs in the appendix seem to be correct, but also mostly follow existing approaches for analyzing IHT (see my comment below). Note that the “subset strong convexity” property (or at least a variation of this property) of random Gaussian matrices seems to have appeared before in the sparse recovery literature; see “A Simple Proof that Random Matrices are Democratic” (2009) by Davenport et al. Couple of questions: - What is \delta in the statement of Lemma 5? - Not entirely clear to me why one would need a 2-stage analysis procedure since the algorithm does not change. Some intuition in the main paper explaining this would be good (and if this two-stage analysis is indeed necessary, then it would add to the novelty of the paper). +++++++++ Update after authors' response +++++++++ Thanks for clarifying some of my questions. I took a closer look at the appendix, and indeed the "fine convergence" analysis of their method is interesting (and quite different from other similar analyses of IHT-style methods). Therefore, I have raised my score.

Reviewer 2



abstract This paper introduces an estimator for robust least-squares regression. First, a comprehensive state of the art is made to recall the existing robust estimators, the computational techniques used and the theoretical guarantees available. Then, the estimator is introduced, motivated by a simple yet smart reformulation of the robust least-squares problem. An iterative hard thresholding approach is used to adjust the model. The obtained estimator is proved consistent and compared to some state-of-the-art competitors in a numerical study. comments I enjoyed reading this paper. I found it well written, with a comprehensive state of the art. Theory of Robust regression is gently introduced, with relevant and not too technical introduction of the statistical guarantees of the existing estimators. The contributions are clearly stated and, to my knowledge, they are important in the field since no robust estimator has been proved consistent in the literature. The hard thresholding approach is well motivated and easy to follow. Simulations are rather convincing. I would have been happy to see, however, some experiments in the high dimensional setting. As a conclusion, I recommend acceptance for this very good manuscript.

Reviewer 3



Summary. This submission studies robust linear regression under the settings where there are additive white observation noises, and where \alpha=\Omega(1) fraction of the responses are corrupted by an oblivious adversary. For this problem, it proposes a consistent estimator, called CRR, which is computationally easy because it is based on iterative hard thresholding. It is shown to be consistent (i.e., converging to the true solution in the limit of the number of responses tending to infinity) under certain conditions (Theorem 4). Quality. In Theorem 4, I did not understand how \epsilon is defined, and consequently, the statement of the theorem itself. Is it defined on the basis of the problem or algorithm setting? Or does the statement of the theorem hold "for any" \epsilon>0, or does "there exist" \epsilon for which the statement hold? The same also applies to \epsilon and e_0 in Lemma 5. In Figure 1, the results with d=500, n=2000, \sigma=2, and k=600 appear in the subfigures (a), (b), and (c), but those in the subfigure (a) seem different from those in (b) and (c), which might question the validity of the results presented here. Clarity. I think that this submission is basically clearly written. Originality. Although I have not done an extensive survey in the literature, proposing a computationally efficient algorithm, with consistency guarantee in the limit of the number n of responses tending to infinity, for the robust linear regression problem with Gaussian noise as well as with oblivious adversary, when the fraction \alpha of adversarial corruptions is \Omega(1). Significance. Even though the fraction \alpha of adversarial corruptions for which CRR is shown to be consistent is quite small (\alpha \le 1/20000), it is still nontrivial to give a consistency guarantee even if a finite fraction of responses are corrupted. It would stimulate further studies regarding finer theoretical analysis as well as algorithmic development. Minor points: Line 64: incl(u)ding more expensive methods Line 76: Among those cite(d) Line 82 and yet guarantee(s) recovery. Line 83: while allowing (allow) Lines 61, 102, and 115: The notation E[...]_2 would require explicit definition. Equation (5): The equation should be terminated by a period rather than a comma. Line 164: (more fine -> finer) analysis Line 173: the (Subset) Strong Smoothness Property Line 246: consequently ensure(s) Lines 254-264: I guess that all the norms appearing in this part should be the 2-norm, but on some occasions the subscript 2 is missing. Lines 285-286: Explicitly mention that the Torrent-FC algorithm was used for comparison. Simply writing "the Torrent algorithm" might be ambiguous, and also in the figures the authors mentioned it as "TORRENT-FC". Line 293: faster runtimes tha(n) Torrent Line 295: both Torrent (and) ex-Lasso become(s) intractable with a(n) order Line 296: Figure(s) 2 (c) and 2(d) show(s) Line 303: Figure 2 (a -> b) shows Line 310: we found CRR to (to) be